# Ice thickness and water level estimation for ice-covered lakes with satellite altimetry waveforms and backscattering coefficients

Xingdong Li[1, 3], Di Long[1, 3], Yanhong Cui[1, 3], Tingxi Liu[2, 3], Jing Lu[4], Mohamed A. Hamouda[5, 6], and Mohamed M. Mohamed[5, 6]

[1]State Key Laboratory of Hydroscience and Engineering, Department of Hydraulic Engineering, Tsinghua University, Beijing 100084, China

[2]Water Conservancy and Civil Engineering College, Inner Mongolia Key Laboratory of Water Resource Protection and Utilization, Inner Mongolia Agricultural University, Hohhot, 010018, China

[3]Collaborative Innovation Center for Integrated Management of Water Resources and Water Environment in the Inner Mongolia Reaches of the Yellow River, Hohhot, 010018, China

[4]State Key Laboratory of Remote Sensing Science, Aerospace Information Research Institute, Chinese Academy of Sciences, Beijing 100101, China

[5]Department of Civil and Environmental Engineering, United Arab Emirates University, Al Ain, 15551, United Arab Emirates

[6]National Water and Energy Center, United Arab Emirates University, Al Ain, 15551, United Arab Emirates

*Correspondence to*: Di Long (dlong@tsinghua.edu.cn) and Tingxi Liu (txliu1966@163.com)

**Abstract.** Lake ice, serving as a sensitive indicator of climate change, is an important regulator of regional hydroclimate and lake ecosystems. For ice-covered lakes, traditional satellite altimetry-based water level estimation is often subject to winter anomalies that are closely related to the thickening of lake ice. Despite recent efforts made in exploiting altimetry data to resolve the two interrelated variables, i.e., lake ice thickness (LIT) and water level of ice-covered lakes, several important issues remain unsolved, including the inability of estimating LIT with altimetric backscattering coefficients in ungauged lakes due to the dependence on in situ LIT data. It is still unclear what role lake surface snow plays in the retrieval of LIT and water levels in ice-covered lakes with altimetry data. Here we developed a novel method to estimate lake ice thickness by combining altimetric waveforms and backscattering coefficients without using in situ LIT data. To overcome complicated initial LIT conditions and better represent thick ice conditions, a logarithmic regression model was developed to transform backscattering coefficients into LIT. We investigated differential impact of lake surface snow on estimating water levels for ice-covered lakes when different threshold retracking methods are used. The developed LIT estimation method, validated against in situ data and cross-validated against modelled LIT shows an accuracy of ~0.2 m and is effective in detecting thin ice that cannot be retrieved by altimetric waveforms. We also improved estimation of water levels for ice-covered lakes with a strategy of merging lake water levels derived from different threshold methods. This study facilitates a better interpretation of satellite altimetry signals from ice-covered lakes and provides opportunities for a wider application of altimetry data to the cryosphere.

**1 Introduction**

Lake ice plays a unique and critical role in regulating lake ecosystems through the modulation of fluxes in and out of the lake, e.g., solar radiation, evaporation, sensible heat, and methane emission (Cooley et al., 2020; Engram et al., 2020; Sharma et al., 2019; Wang et al., 2018; Wik et al., 2016; Woolway et al., 2020). The vulnerability of lake ice to climate change causes wide concern to the stability of boreal lake ecosystems and the sustainability of socioeconomic activities that rely on lake ice (Knoll et al., 2019; Mullan et al., 2017). Lake ice cover and LIT are two Essential Climate Variables (ECVs) related to lake ice identified by the Global Climate Observing System (GCOS). Lake ice cover is a measure of lake ice quantity (horizontally). LIT can provide information on both lake ice quantity (vertically) and quality (e.g., the strength of lake ice), which is highly related to the safety of human activities on ice. For instance, LIT loss could reduce the availability of ice roads (Li et al., 2022a) and increase the possibility of winter drowning (Sharma et al., 2020). However, compared with the intensively investigated lake/river ice cover (Du et al., 2017; Yang et al., 2020; Kropácek et al., 2013), the knowledge of LIT is largely limited, due mostly to the lack of in situ observations and effective remote sensing-based methods. There is a considerable gap between the monitoring accuracy of LIT expected by the GCOS (1–2 cm) and that of current remote sensing-based approaches (0.1–0.2 m). For winter water level estimation based on altimeters, the existence of lake ice is a barrier that could cause an abrupt decrease in altimetric lake surface height (LSH) (Shu et al., 2020). To resolve this issue, a better understanding of the impact of lake ice and lake surface snow on altimetric signals is necessary.

Current remote sensing of LIT is based mostly on information from thermal infrared sensors and microwave sensors (Murfitt and Duguay, 2021). Thermal infrared information such as lake surface temperatures can be used to drive a freezing degree day-based model or more sophisticated lake ice models to estimate LIT (Yu and Rothrock, 1996; Pour et al., 2017; Zeng et al., 2016; Li et al., 2022a). However, cloud contamination and complex physical processes related to lake surface snow (Cheng et al., 2013; Duguay et al., 2003) could limit the accuracy and robustness of the method based on thermal infrared information and lake ice modelling. Microwave information has a certain penetration depth (Atwood et al., 2015) within the lake ice and is not affected by cloud cover, providing great potential of more direct and robust observations of LIT.

Some previous studies focused on the use of passive microwave information, i.e., brightness temperature ($T_B$) obtained by satellite radiometers. Kang et al. (2010) explored the relationship between $T_B$ obtained by AMSR-E and LIT in two Canadian lakes, Great Slave Lake (GSL) and Great Bear Lake (GBL), indicating that the increase in LIT is associated with the increase in $T_B$. They later showed that, with a linear regression model, an 18.7 GHz $T_B$ could best represent the LIT accumulation and the accuracy (root mean squared error, RMSE) was ~0.18 m (Kang et al., 2014). Passive microwave methods perform well in terms of high temporal resolution (daily) but are limited to a few large lakes due to the low spatial resolution, as the pixel size of the 18.7 GHz $T_B$ is 25 km.

Active microwave remote sensing of LIT can be further categorized into classes based on: (1) SAR images or (2) satellite altimetry. Backscattering coefficients of SAR images would experience a rapid decrease when the lake surface is covered by

skim ice (a quasi-specular reflector), followed by a steady increase with the accumulation of LIT until the floating lake ice becomes bedfast lake ice or the melting starts (Duguay and Lafleur, 2003; Murfitt and Duguay, 2021; Howell et al., 2009a; Murfitt et al., 2018). Given the mentioned behaviours, backscattering coefficients from SAR images were widely used in discriminating bedfast lake ice from floating lake ice and monitoring of lake/sea ice phenology (Howell et al., 2018; Howell et al., 2019). However, due to the high variability in roughness associated with ice growth, SAR image-based LIT estimation is subject to larger uncertainty when the ice thickness exceeds 40 cm (Murfitt and Duguay, 2021).

Satellite altimeters were initially designed for monitoring ocean topography. Nevertheless, numerous studies have explored the potential of satellite altimetry in monitoring inland waters such as river water levels and discharge, lake water levels and storage changes, glacier elevation changes and mass balance, and recently in LIT (Huang et al., 2019; Zakharova et al., 2021; Murfitt et al., 2022; Beckers et al., 2017; Zhang et al., 2021; Zhao et al., 2022; Li et al., 2022a; Huang et al., 2018; Li et al., 2019). Altimetry-based LIT can be derived from backscattering coefficients or radar waveforms. Different from SAR images indicated above, backscattering coefficients from satellite altimeters would experience a rapid increase when the open water is covered with skim ice, followed by a steady decrease with the thickening of LIT until the melting starts. A recent study (Zakharova et al., 2021) investigated the relationship between the altimetry-based backscattering coefficients and in situ river ice thickness, suggesting the great potential of altimetry-based backscattering coefficients in estimating LIT for thin ice. However, in situ ice thickness data are necessary to derive regression models, which greatly limits applications of the method developed by Zakharova et al. (2021). To avoid confusion, the term "backscattering coefficients" refers to altimetry-based backscattering coefficients in the following context, unless otherwise stated.

LIT estimation based on satellite altimetric waveforms was first investigated by Beckers et al. (2017) with double-peak waveforms from CryoSat-2 on GSL and GBL, which provides a potential approach for robust LIT monitoring because the method is physically-based and does not rely on parameterization. Shu et al. (2020) combined the method developed by Beckers et al. (2017) in winter water level retrieval using Sentinel-3 data. CryoSat-2 and Sentinel-3 are SAR altimeters with pulse-doppler-limited footprints, which can be regarded as beam-limited footprints. Compared with traditional pulse-limited altimeters such as TOPEX/Poseidon (T/P) and Jason-1/2/3 (available since 1992), the time span of SAR altimeters such as CryoSat-2 and Sentinel-3 is relatively short (i.e., CryoSat-2 was launched in 2010 and Sentinel-3A was launched in 2016). The method developed by Beckers et al. (2017) is not that compatible with traditional pulse-limited altimeters, because the waveforms of pulse-limited altimeters are largely different from those from SAR altimeters. Li et al. (2022a) developed a LIT estimation method suitable for pulse-limited altimeters T/P and Jason-1/2/3. Therefore, the time span of retrievable LIT has been increased substantially from ~10 years to almost three decades. The temporal resolution has also been largely improved because T/P and Jason-1/2/3 have the shortest revisit cycle (~10 days) among all existing satellite altimeters. However, the LIT estimation for thin ice based on radar waveforms is limited by the range resolution of the waveform. For instance, the minimum LIT retrievable with the method developed by Beckers et al. (2017) is 0.263 m for CryoSat-2 theoretically. For

Jason-1/2/3, Li et al. (2022a) suggested that the LIT retrieval is robust after LIT exceeds 0.4 m because the waveforms of Jason-1/2/3 have a coarser range resolution than CryoSat-2.

Water level estimation for ice-covered lakes is essential for water resources management in cryosphere under a changing climate (Li et al., 2022b; Long and Li, 2022; Wu et al., 2022) and has been investigated with different approaches for different altimeters (Shu et al., 2020; Yang et al., 2021; Ziyad et al., 2020). Ziyad et al. (2020) developed a classification scheme to

separate Jason-2 observations from the ice-covered lake surface from the open water surface, and only used open water observations to derive water level time series to avoid the contamination from lake ice. Shu et al. (2020) applied the method developed by Beckers et al. (2017) to estimate LIT using Sentinel-3, and then derived a range correction associated with LIT to correct the abrupt drop in winter altimetric water levels. Yang et al. (2021) tested several threshold retracking algorithms to develop a modified subwaveform threshold (MST) retracking method for two-peak waveforms from T/P and Jason-1/2/3 to

improve water level estimation during ice seasons. The MST retracking algorithm could avoid winter water level anomalies for most cases and the metrics of derived altimetric water levels are quite promising, e.g., the standard deviations (STDs) of the differences between altimetric water levels and in situ water levels are mostly smaller than 0.1 m among study lakes (GSL, GBL, and Athabasca Lake). However, an important issue remains to be further discussed. Causes of the two-peak waveforms are still not clear and could be attributed to multiple backscattering surfaces, i.e., snow surface, snow-ice interface, and ice-

water interface. Yang et al. (2021) suggested that the first subwaveform of Jason-1/2/3 waveforms from ice-covered lake surfaces corresponds to snow-ice interfaces based on the comparison with in situ water levels. However, Li et al. (2022a) suggested that the first subwaveform corresponds to the snow surface for most Canadian lakes based on the comparison with in situ ice and snow thickness. Better understanding the formation of altimetry radar waveforms from ice-covered lake surfaces could benefit the retrieval of winter water levels as well as LIT.

This study was designed to: (1) combine satellite altimetry-based waveforms and backscattering coefficients to improve LIT estimation for ungauged lakes and thin ice, and (2) explore possible improvements in altimetric water level estimation for ice-covered lakes through a better understanding of altimetric signals from snow and ice-covered lake surfaces. As mentioned above, LIT estimation based on waveforms alone is ineffective for thin ice and altimetry-based backscattering coefficients have the potential to monitor thin ice. Meanwhile, the dependence on in situ data limits a wider application of altimetry-based

backscattering coefficients to LIT estimation. Therefore, the combination of these two methods (satellite altimetry-based backscattering coefficients and waveforms) could be complementary. To exploit the potential of backscattering coefficients in LIT estimation, we derived a logarithmic regression model to better represent various lake ice conditions, which is detailed in Sect. 3.2 and Sect. 4.1. As for water level estimation, we mainly explored different behaviours of lake surface snow when different threshold methods were used. We then developed an approach of merging water level time series derived from

different threshold methods.

This paper is organized as follows. Sect. 2 introduces the study area and data used. Sect. 3 provides details on LIT estimation based on the combination of backscattering coefficients and waveforms from satellite altimetry, as well as an improved water

level estimation method for ice covered lakes. Sect. 3 also includes a detailed deduction of an original logarithmic regression model used to convert backscattering coefficients into LIT. Sect. 4 shows the performance of the logarithmic model and the

validation of LIT and water level estimation methods. Sect. 5 discusses differential impact of lake surface snow when different threshold methods are used, uncertainty sources of LIT estimation and water level retrieval, and implications of this study in future lake ice and lake water level research. Sect. 6 summarizes the main findings of this study.

## 2 Study area and data

### 2.1 Study area

As shown in Fig 1, we investigated eight lakes, including five lakes in Canada, i.e., GBL (121.30°W, 65.91°N), GSL (114.37°W, 62.09°N), Athabasca Lake (109.96°W, 59.10°N), Winnipeg Lake (97.25°W, 52.12°N), and Baker Lake (95.28°W, 64.13°N), and two lakes in Asia, i.e., Hulun Lake (117.38°E, 48.97°N) and Har Lake (93.21°E, 48.05°N), and one lake in Europe, i.e., Peipus Lake (27.45°E, 58.65°N). Environmental and climatic conditions of the study lakes are summarized in Table 1. GBL, GSL, and Athabasca Lake are located in the Mackenzie River basin, where mean annual temperature ranges

from -10 – 3°C from the northern to the southern part of the basin. Mean annual precipitation in the Mackenzie River basin is 410 mm but ranges between 300 mm and 1,000 mm from northeast to southwest (Howell et al., 2009a; Abdul Aziz and Burn, 2006). Baker Lake is located in the northeast part of Canada, with an annual air temperature of -9.6°C and an annual precipitation of 157 mm (Medeiros et al., 2012). Winnipeg Lake covers a wide range of latitudes and mean annual air temperatures vary considerably from south (1.6°C) to north (-0.7°C). Mean annual precipitation in the Winnipeg Lake basin

is 498 mm (Stewardship, 2011). Hulun Lake has an annual temperature of 2.3°C and an annual precipitation of 240 mm that mostly takes place from June to September due to a continental monsoon climate (Cai et al., 2016; Wu et al., 2019). Har Lake (Khar Lake) is located in a desert in Mongolia, with an annual temperature of ~ 0.8°C and an annual precipitation of ~ 50 mm based on reanalysis data and a surface water resource report (https://raise.suiri.tsukuba.ac.jp/new/press/youshi_sugita8.pdf). Peipus Lake is lying on the border of Russia and Estonia, with a mean annual temperature of ~ 6°C and a mean annual

precipitation of ~ 630 mm based on climate records from the Estonia Environment Agency (https://www.ilmateenistus.ee/?lang=en). Among the eight study lakes, based on the availability of in situ measurements and environmental conditions, GSL, Baker Lake, Peipus Lake, Hulun Lake, and Har Lake were selected for testing the LIT retrieval method, while GSL, GBL, Athabasca Lake, and Winnipeg Lake were selected for the test of water level estimation method.

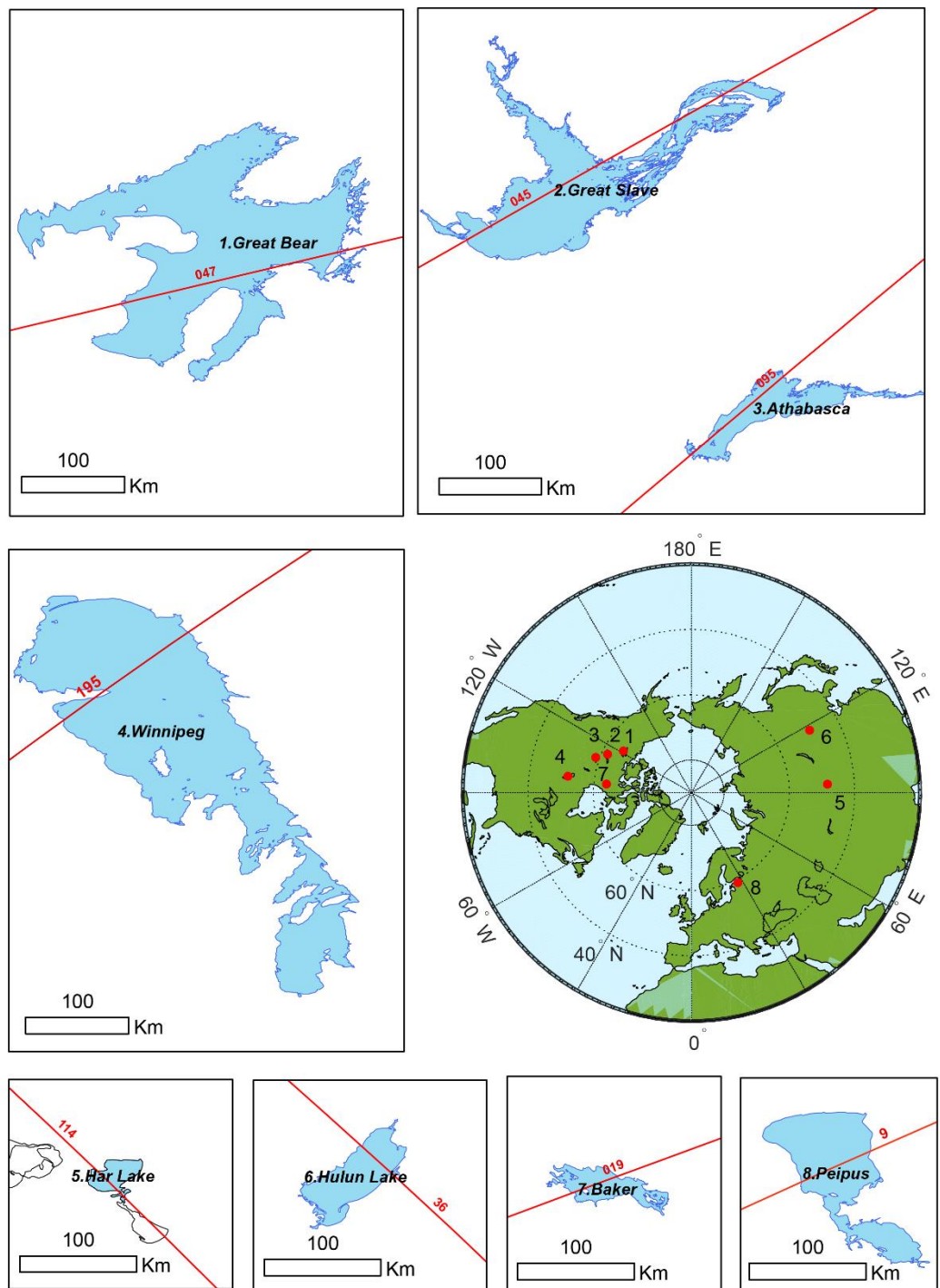

**Figure 1: Study lakes and satellite altimetry ground tracks used. Red curves denote ground tracks of T/P and Jason-1/2/3. Red numbers denote the ground track number.**

**Table 1 Environmental and climate conditions of study lakes**

| Lake/region name | Mean air temperature (℃) | Winter Air temperature (℃) | Precipitation (mm) | Location | Reference |
|---|---|---|---|---|---|
| Mackenzie River basin (GBL, GSL, Athabasca Lake) | -10 – 3 | -35 – -25 | 410 | ~115 °W ~62 °N | (Abdul Aziz and Burn, 2006; Howell et al., 2009b) |
| Baker Lake | -9.6 | -30 – -20 | 157 | 95.28°W 64.13°N | climate.weather.gc.ca and Medeiros et al. (2012) |
| Winnipeg Lake | -0.7 – 1.6 | -20 – -5 | 498 | 97.25°W 52.12°N | climate.weather.gc.ca and Stewardship (2011) |
| Hulun Lake | 2.3 | -16 – -10 | 240 | 117.38°E 48.97°N | (Wu et al., 2019) and (Wang et al., 2017) |
| Har Lake | ~0.8 | -15 – -5 | ~50 | 93.21°E 48.05°N | Mongolia surface water report and reanalysis data |
| Peipus Lake | ~6 | -5 – -2 | 630 | 27.45°E, 58.65°N | www.ilmateenistus.ee |

**2.2 Data**

Satellite altimetry data we used here were collected by Jason-1/2/3, covering the 2002–2020 period. Ground tracks for each lake are shown in Fig. 1. Jason-1/2/3 are follow-on missions of T/P and inherited the orbit of their predecessor. T/P and Jason-1/2/3 have the shortest revisit time of ~10 days among existing satellite altimetry missions, providing observations from 66 °N to 66 °S. Radar altimeters carried by Jason-1/2/3 are dual-frequency (Ku-band and C-band) pulse-limited altimeters. Pulse-limited essentially means that the size of radar altimetry illuminated area/footprints is limited by the pulse width, as opposed
to the beam width (such as laser altimeters and SAR altimeters). As a result, the trailing edge of pulse-limited waveforms is milder and noisier than that of beam-limited waveforms, adding to the difficulty of retrieving LIT based on waveforms.

Altimetry products used here were the Sensor Geophysical Data Records (SGDR), containing waveforms, backscattering coefficients for Ku-band and C-band, satellite altitude, uncorrected range, and range corrections (atmospheric corrections and geophysical corrections) for 20 Hz footprints (20 footprints per second, with a spacing of ~330 m). The SGDR products also
provide corrected ranges using several retracking algorithms (ICE, MLE3, and MLE4), but have been shown unreliable in water level estimation for ice-covered lakes (Yang et al., 2021). However, it does not mean that default retracking algorithms (MLE4) are irrelevant to this study. On the contrary, backscattering coefficients provided in the SGDR products are generated

from the MLE4 retracking algorithm and are highly related to the amplitude of the waveforms. The altimetry data used can be obtained from the Archiving, Validation, and Interpretation of Satellite Oceanographic (AVISO+) (http://ftp-access.aviso.altimetry.fr).

To validate the derived LIT, we obtained in situ LIT for GSL and Baker Lake collected by the Ice Thickness Program Collection, which is available at (https://www.canada.ca/en/services/environment/weather/other-services.html). The data set contains weekly in situ snow and ice thickness measured with drilled holes. We also obtained in situ LIT of Lake Peipus from hydrological yearbooks of Estonia, which is available at (https://www.ilmateenistus.ee/?lang=en). Sampling positions of GSL, Baker Lake, and Peipus Lake are listed in Table 2. It should be noted that in situ ice thickness data are often measured near the shore, where the lake water freezes earlier and the ice thickness could be larger at the beginning of ice seasons (Murfitt et al., 2022; Mangilli et al., 2022). Data records for GSL and Baker Lake have been updated to 2016 and 2020, respectively. To validate the derived altimetric water levels, we obtained daily gauge water levels for GBL, GSL, Athabasca Lake, and Winnipeg Lake collected by the Water Survey of Canada, available at (https://wateroffice.ec.gc.ca/index_e.html). Gauge station names, station codes, locations, and record time span for different lakes are listed in Table 2. The in situ water levels were measured with pressure sensors and therefore represent the free water surface (Yang et al., 2021). Given that in situ and altimetric water levels are based on different datums, we removed the systematic bias between them before making any comparison. The systematic bias is defined as the mean difference between in situ water level time series and altimetric water level time series.

**Table 2 In situ lake water level gauging stations and LIT observation sites used in this study**

| Lake Name | Data | Station ID | Location | Available records time |
|-----------|------|-----------|----------|------------------------|
| Great Bear Lake | Water level | 10JE002 | 66.6°N, 117.6°W | 1984/7/10–2018/12/31 |
| Great Slave Lake | Water level | 07OB001 | 62.4°N, 114.35°W | 1934/1/31–2018/12/31 |
| | Ice Thickness | / | 62.4°N, 114.35°W | 1992/1/2–2016/4/28 |
| Winnipeg Lake | Water level | 05SG001 | 53.18°N, 99.2°W | 1953/6/24–2020/12/31 |
| Athabasca Lake | Water level | 07MC003 | 59.38°N, 108.88°W | 1956/2/25–2020/12/31 |
| Baker Lake | Ice thickness | / | 64.3°N, 96.0°W | 1992/1/10–2021/12/31 |
| Peipus Lake | Ice thickness | / | 58.83°N, 26.99°E | 1991/1/8–2015/2/25 |

We also used modelled LIT to provide cross-validation for lakes without in situ LIT measurements. The modelled LIT data were based on a one dimensional remote sensing lake ice model developed by Li et al. (2022a). The 1-D lake ice model developed by Li et al. (2022) has a similar structure as the HIGHTSI model, but it uses MODIS sensor LST as the upper boundary condition to solve the heat transfer equation within lake ice and surface snow. MODIS albedo was also incorporated

to reduce uncertainty in simulated surface snow depth. Based on validation against in situ data (e.g., in Baker Lake, GSL, and Pepsi Lake), the remote sensing lake ice model shows an accuracy of 0.1–0.2 m (RMSE). The modelled LIT is online available at https://doi.org/10.5281/zenodo.5528542.

## 3 Method

A workflow of the methods to retrieve LIT and LSH is illustrated in Fig. 2. Ku-band backscattering coefficients of Jason-1/2/3

were first extracted to classify the type of the observation, i.e., the open water period and the ice-covered period. The LIT would first be estimated based on double-peak waveforms to be illustrated in Sect. 3.1. Then the initial LIT results were used to derive a regression model with Ku-band backscattering coefficients of Jason-1/2/3 to transform backscattering coefficients into LIT to be explained in Sect. 3.2. Subsequently, the LITs based on waveforms and backscattering coefficients were merged and validated/cross-validated against in situ LIT/modelled LIT.

LSH estimation is based on threshold retracking methods to be detailed in Sect. 3.3. For the double-peak waveform in the ice-covered period, only the first subwavefrom is used to retrieve LSH, which is similar to what Yang et al. (2021) have done. Here we used different thresholds (0.1 and 0.5) to generate two LSH time series (LSH_01 and LSH_05 in Fig. 2), because they have different performance in open water and ice-covered periods, i.e., the time series derived from a 0.1 threshold could better reveal the LSH for the ice-covered period, whereas that derived from a 0.5 threshold could better represent the LSH for the

open water period. The systematic bias between the two time series based on the 0.1 and 0.5 thresholds during the open water period was removed to merge them into the final LSH time series that were validated with in situ water levels.

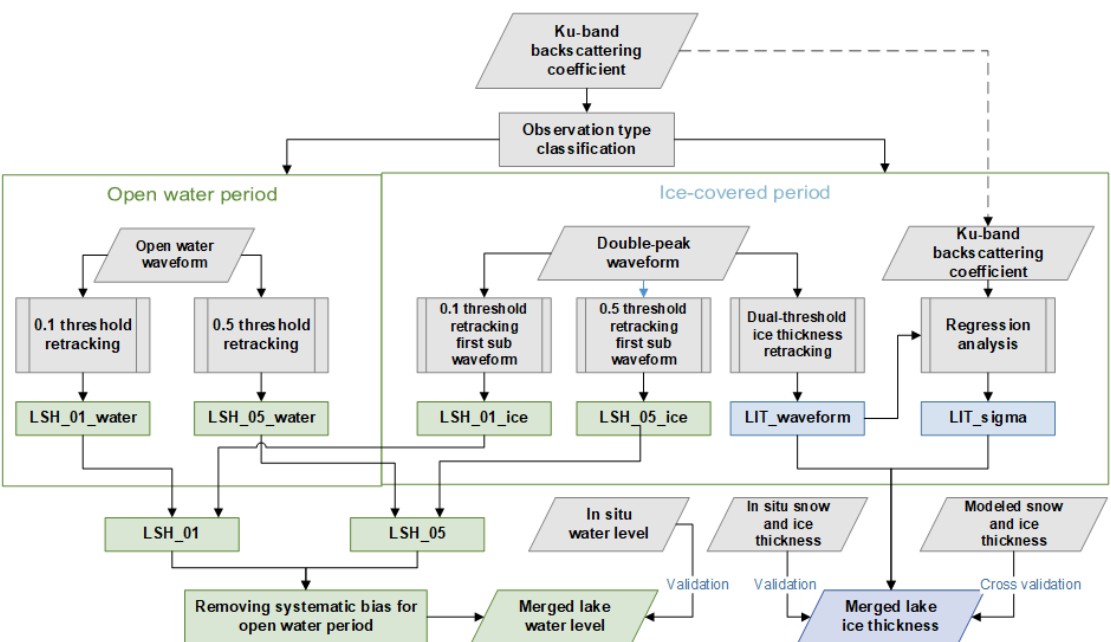

**Figure 2: Workflow of this study.** Procedures/intermediate data associated with LSH estimation are marked with green. Procedures/intermediate data associated with LIT estimation are marked with blue. LSH_01_water denotes that this intermediate data are lake surface height derived with a 0.1 threshold during the open water period. LSH_01_ice denotes the same meaning but for the ice-covered period. LSH_01 denotes the time series containing all LSH retrievals derived with a 0.1 threshold. Similarly, LSH_05_water, LSH_05_ice, and LSH_05 denote LSHs for different periods derived with a 0.5 threshold. LIT_waveform denotes the LIT derived from double-peak waveforms. LIT_sigma denotes LIT derived from backscattering coefficients.

## 3.1 LIT retrieval with satellite altimetry waveforms

LIT estimation based on Jason-1/2/3 waveforms was developed by Li et al. (2022a). Here we provide the basic concepts and steps of this method and some comparisons with previous studies. Altimetry radar waveforms represent the returned radar power as a function of time. When the lake surface is covered with ice and snow, the radar pulse can be backscattered from the air-snow interface, the snow-ice interface, or the ice-water interface. The coupling of signals backscattered from different interfaces could result in the double-peak waveforms. The second peak of the waveform, often highest, is related to the signal from the ice-water interface. However, the source of the first peak is still not clear. The time lag between the two peaks is the time radar pulse transfers between the two interfaces and can be used to calculate the thickness of the medium. Beckers et al. (2017) tested the first peak and the highest peak of CryoSat-2 waveforms to estimate the LIT. SAR altimeters such as CryoSat-2 can be seen as the beam-limited in the along-track direction. The beam-limited waveforms have steep trailing edges so that it is easier to identify peaks associated with the ice-water interface. Nonetheless, for pulse-limited altimeters with a mild and noisy trailing edge, multiple peaks in the trailing edge are not uncommon, as shown in Fig. 3 (a), making it difficult to select

the correct peak associated with the ice-water interface. In addition, LITs derived using waveform peaks are discrete because the time difference between different peaks is multiple integers of a bin width (3.125 ns for Jason-1/2/3). Therefore, Li et al. (2022a) developed a dual-threshold retracking algorithm to estimate the LIT with Jason-1/2/3 waveforms.

Procedures of the dual-threshold retracking method are as follows.

(1) Find the inflection point $T$ on the leading edge of the waveform. If the inflection point appears near the middle of the leading edge, it indicates that there could be two peaks on the leading edge representing the snow/ice surface and ice bottom. If the inflection point appears close to the top of the leading edge, it suggests that there is only one peak on the leading edge and the waveform will be discarded. Assume that the waveform is comprised of $P_1$, $P_2$ ... $P_N$. The power
difference for adjacent bins can be calculated as $D_i = P_{i+1}-P_i$. $S$ is the STD of $D_1$, $D_2$ ... $D_{N-1}$. The first bin of the leading edge is defined as the G0, which satisfies $D_{G0} > 0.2{\times}S$. Then [G0, G0+15] is defined as the search window. The inflection point $T$ in the search window satisfies $D_T < D_{T-1}$. Subsequently, find the maximum power $P_M$ in the search window. If $P_T > 0.9P_M$, discard the waveform, because the inflection point appears near the top of the leading edge.

(2) The first subwaveform associated with the snow/ice surface is defined as [$P_{G0}$, … $P_{T+1}$], while the second subwaveform
associated with the ice bottom is defined as [$P_T$, … $P_M$]. Then, two thresholds ($Th_1$ and $Th_2$) can be calculated to determine the two tracking points for the snow/ice surface and ice bottom. Here we used a 0.5 threshold to calculate $Th_1$ and $Th_2$, as shown by Equations (1–2). The two tracking points ($T1$ and $T2$) can be calculated with Equations (3–4).

$$Th_1 = 0.5 \times (P_{G0} + P_{T+1}) \tag{1}$$

$$Th_2 = 0.5 \times (P_T + P_M) \tag{2}$$

$$T1 = x + \frac{Th_1-P_x}{P_{x+1}-P_x}, where\ P_x < Th_1, P_{x+1} > Th_1 \tag{3}$$

$$T2 = y + \frac{Th_2-P_y}{P_{y+1}-P_y}, where\ P_y < Th_2, P_{y+1} > Th_2 \tag{4}$$

The ice thickness can be calculated as $LIT = 0.5{\times}(T2-T1){\times}c_i{\times}3.125{\times}10^{-9}$, where $c_i$ is the speed of microwave in ice. The $c_i$ is calculated with $c/n_i$, where $n_i$ is 1.78, the refractive index of ice at the Ku-band (Warren and Brandt, 2008). After acquiring LITs for all footprints for each cycle, the median LIT of each cycle will be used to form the LIT time series.

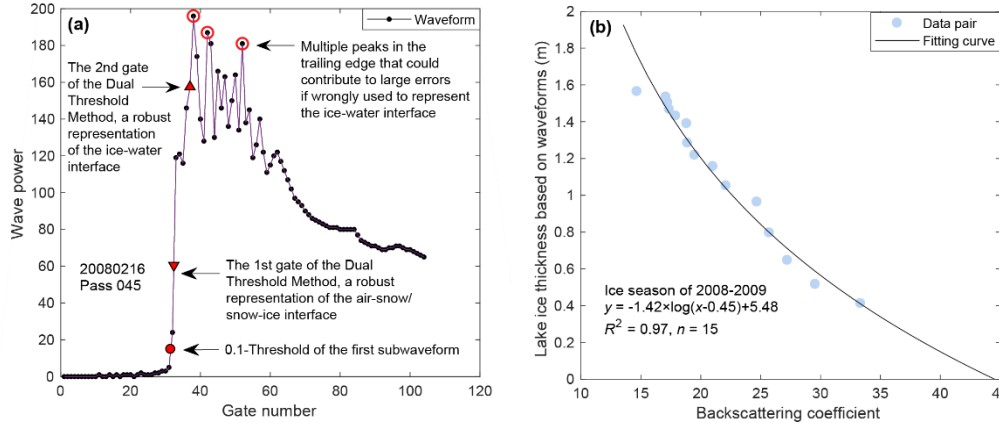


**Figure 3: Mechanisms of LIT estimation based on waveforms and backscattering coefficients. (a) a Jason-2 waveform obtained from ice-covered GSL. Red triangles indicate retracking points derived from the dual-threshold retracking algorithm, which can be used to estimate LIT. The red solid cycle denotes the retracking point of the first subwaveform with a 0.1 threshold retracker, which can be used to derive LSH. (b) Scatterplot of backscattering coefficients and LIT derived from the dual-threshold retracking algorithm**

**through the ice season of 2008–2009 for GSL. Backscattering coefficients can be transformed into LIT with the derived regression model.**

### 3.2 LIT retrieval with satellite altimetry backscattering coefficients

Evolution in backscattering coefficients during ice seasons is complicated and very different between satellite altimetry and

SAR images. For open water, altimetric backscattering coefficients are relatively low, ranging from 10–20 dB for different cycles. For the same cycle, the spatial variation of altimetric backscattering coefficients on the open water is small (< 1 dB), as shown by the red curve in Fig. 4(b). Overall, there are four stages of variations in backscattering coefficients with ice evolution during ice seasons (dashed line boxes in Fig. 4(a)). Stage I refers to the period when the lake starts to freeze and be covered by skim ice. During this stage, the altimetric backscattering coefficients soar to a high value in a year (e.g., the first

dashed line box in Fig. 4(a)), which could be attributed to the quasi-specular reflecting effect of the smoothed lake surface. Meanwhile, the spatial variation of the altimetric backscattering coefficients becomes relatively large (generally > 2 dB), as shown by the blue curve in Fig. 4(c).

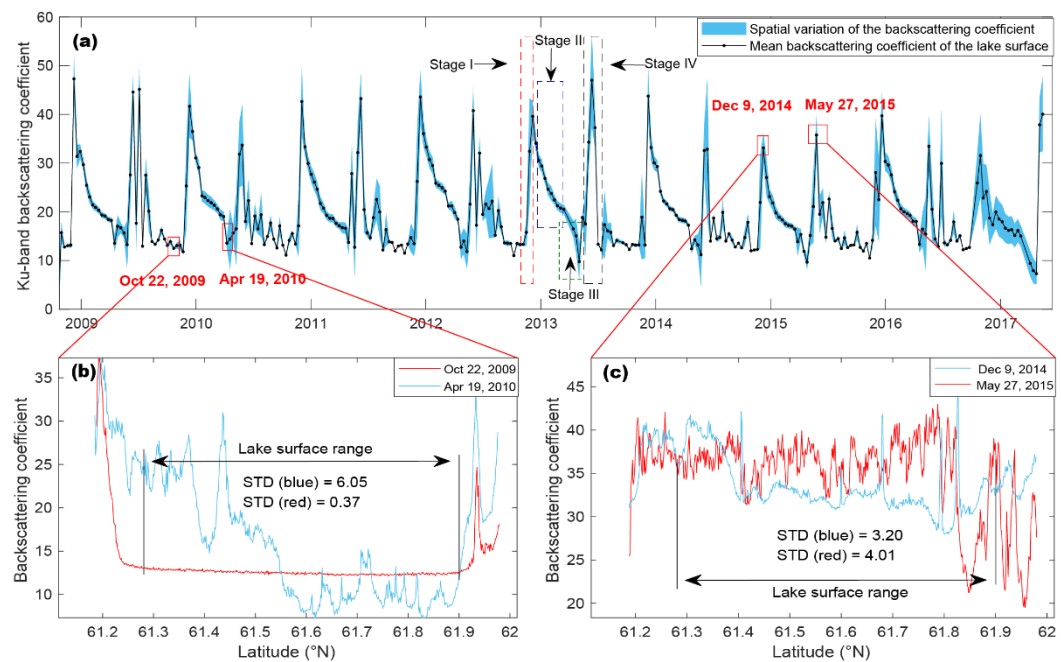

**Figure 4: Temporal and spatial variations of Jason-2 Ku-band backscattering coefficients on GSL. (a) Time series of mean backscattering coefficients for each cycle during 2009–2017. Blue shading areas denote the STD of backscattering coefficients for each cycle. (b) and (c) are distributions of the backscattering coefficients along with latitudes for specific cycles/dates (including Oct 22, 2009 and Apr 19, 2010 in (b), and Dec 09, 2014 and May 27, 2015 in (b)). Lake surface latitudinal ranges are marked with double-arrows in (b) and (c).**

In comparison, backscattering coefficients derived from SAR images experience high and low values due to wind-induced lake surface roughness for open water periods (Horstmann et al., 2003; Horstmann et al., 2000), but decrease rapidly when the lake starts to freeze. The reason why the altimetric backscattering coefficients deviate from those based on SAR images is that altimetry data are nadir-looking observations while most pixels in SAR images are side-looking observations (Fu and Cazenave, 2000; Peureux et al., 2022). Consequently, the incident direction is collinear (noncollinear) with the reflection direction for satellite altimetry (SAR images). Therefore, the backscattered energy is high (low) by the quasi-specular reflector for satellite altimeters (SAR images). In the following context, the term "backscattering coefficients" refers to altimetry-based backscattering coefficients.

During stage II, with the increase in LIT, the backscattering coefficients start to decrease steadily until the melting starts (e.g., the second dashed line box in Fig.4(a)). The decrease in backscattering coefficients could be attributed to the increased absorption and volume scattering associated with the increased LIT. During stage III when the melting begins, there will be an abrupt decrease in backscattering coefficients (e.g., the third dashed line box in Fig. 4(a)), which is caused partially by ice

metamorphism (formation of dendroidal air channels just below the ice surface and early stages of needle ice formation) (Kouraev et al., 2015). As shown by the blue curve in Fig. 4(b), the backscattering coefficients are very low (e.g., < 10 dB) and noisy over the melting lake surface. The STD of backscattering coefficients here representing the variability in the spatial domain for the melting period is much larger than that for the open water period. Based on this phenomenon, we set a criterion (the mean backscattering coefficients < 15 dB and STD > 1.5 dB) to filter out the abrupt decrease in backscattering coefficients when the melting starts, because these low values would lead to unrealistic large LIT estimates.

During stage IV, as the LIT continues to decrease with the melting process, backscattering coefficients start to increase to a high value once again, because of the decrease in absorption and volume scattering effect. Eventually, once the ice completely melts, the backscattering coefficients drop to a level of the open water surface (e.g., the fourth dashed line box in Fig. 4(a)). Therefore, the highest peak in the freezing period and the highest peak in the melting period were selected to characterize the ice-on and ice-off dates, which classifies the observations into either open water observations or ice-covered observations as suggested by Zakharova et al. (2021). We compared the backscattered ice-on and ice-off dates against ice phenology manually identified with MODIS images in GSL, finding high agreement between the two independent data (Supplementary Figure S1).

Based on the variability in backscattering coefficients during the ice seasons, Zakharova et al. (2021) assumed the decrease in backscatter between two consecutive observations to be proportional to the gain in ice thickness and derived a regression model between the cumulative backscatter difference and the in situ river ice thickness on the Lower Ob River. The regression model has the form of $H_i = a \times \text{CumSum}(\text{d}Sig/\text{d}t)^b$, where $H_i$ is the ice thickness, $\text{CumSum}(\text{d}Sig/\text{d}t)$ is the cumulative backscatter difference, and $a$ and $b$ are model parameters calibrated against in situ ice thickness. For simplicity, this model is referred to as the power function model in the following context. The power function model does not consider the physical process associated with ice growth and is dependent on in situ measurements, which limits a wider application of the method. In addition, we found that the performance of LIT estimation using only one regression model with one set of model parameters can be fairly unstable from year to year and from lake to lake, partially because the initial ice and snow conditions can be very different for each winter and each lake, which is also mentioned by Zakharova et al. (2021).

We developed a new regression model considering the physical processes and applied this model to relate backscattering coefficients with LITs derived from waveforms for each lake and during each winter. Therefore, we can circumvent the problems caused by the difference in initial ice and snow conditions. Meanwhile, we can derive LITs based on backscattering coefficients without in situ ice thickness measurements. Because our model has a logarithmic form (equation (10)), it is referred to as the logarithmic model. As shown in Sect. 4.1, the logarithmic model can better represent the LIT compared with the power function model for lakes with thick ice (e.g., > 1 m) and rapid ice accumulation rates.

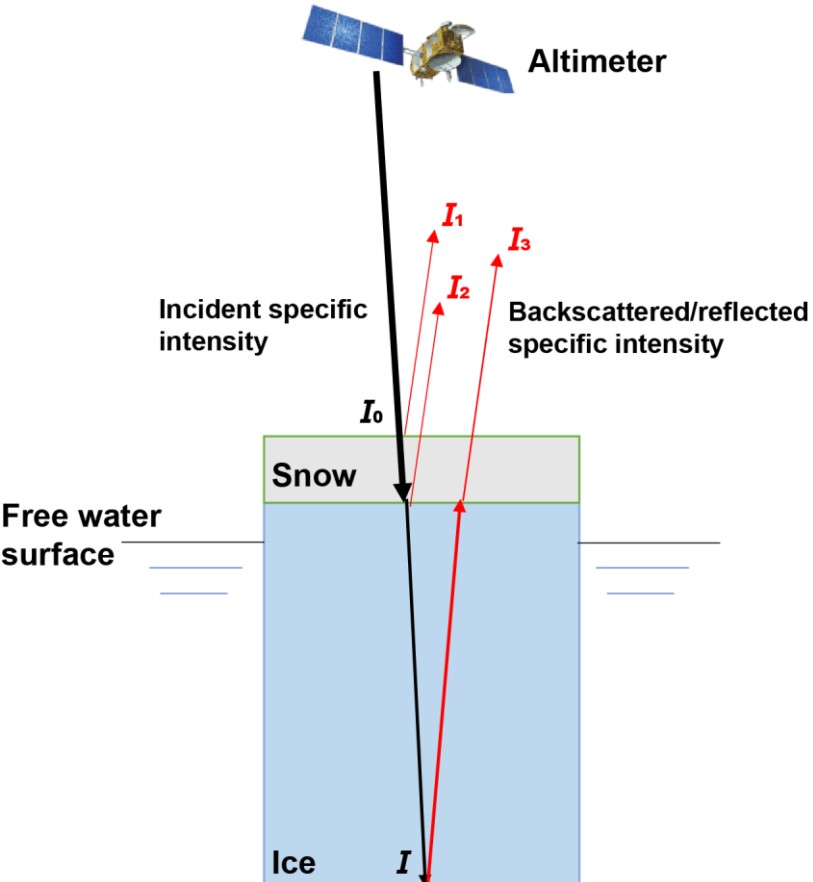

**Figure 5: A schematic diagram of radar altimetry specific intensity backscattered/reflected from ice-covered lakes. Black arrows denote the incident specific intensity. Red arrows denote backscattered or reflected specific intensity. $I_0$ denotes the transmitted microwave intensity just below the snow-ice interface. $I$ denotes the transmitted microwave intensity that has just reached the ice-water interface. $I_1$ denotes the backscattered intensity from the air-snow interface, $I_2$ denotes the backscattered intensity from the snow-ice interface, and $I_3$ denotes the backscattered intensity from the ice-water interface. Note that at the nadir the incident angle is small and the backscattered/reflected pulse is approximately colinear with the incident radar pulse.**

Theoretically, radar pulse would be backscattered from multiple snow and ice layers, given that different snow/ice layers have different density and temperatures that could influence the backscattering process. The backscattered intensity is a function of the distance, direction, and time that requires detailed modelling, as was done by Larue et al. (2021). To provide a straightforward derivation of the regression model we developed, here we focus on the backscattered intensity of the nadir and assume the radar pulse to be backscattered mostly from three interfaces, i.e., the air-snow interface, the snow-ice interface, and the ice-water interface. Volume scattering from snow layers also affects the backscattered intensity, which is discussed in Sect. 5.2. Here we approximate the backscattered intensity $I_b$ with the sum of $I_1$, $I_2$, and $I_3$, i.e., the backscattered intensity from the

air-snow, the snow-ice, and the ice-water interface as shown in Fig. 5. We assume the reflectance for these interfaces to be $R_1$, $R_2$, and $R_3$, respectively. Given the incident intensity $I_0$, the backscattered intensity $I_b$ can be written as Equation (5):

$$I_b = I_1 + I_2 + I_3 = R_1 I_0 + I_2 + I_3 \tag{5}$$

At stage II shown in Fig 4, backscattering coefficients decrease with the increase in LIT, which could be caused by the increased absorption and volume scattering. Here we approximate the extinction of microwave intensity in snow and ice with an exponential equation. For instance, $I_2$ and $I_3$ can be written as Equation (6) and Equation (7):

$$I_2 = R_2(1 - R_1)I_0 \times e^{-2kH_s} \tag{6}$$

$$I_3 = R_3(1 - R_2)(1 - R_1)I_0 \times e^{-2k(H_s + H_i)} \tag{7}$$

where $k$ is an effective extinction coefficient for snow and ice, $H_s$ and $H_i$ denote the thickness of snow and ice, and $R_2$ and $R_3$ denote reflectance at the snow-ice and the ice-water interface.

Note that the backscattered intensity travels a round-trip in the snow and ice. Therefore, the exponential term is written as exp $(-2k (H_s + H_i))$. By substitution of $I_2$ and $I_3$ in Equation (5), the total microwave intensity backscattered from the ice-covered lake surface can be approximated by Equation (8).

$$I_b = (R_1 I_0 + R_2(1 - R_1)I_0 \times e^{-2kH_s}) + R_3(1 - R_2)(1 - R_1)I_0 \times e^{-2k(H_s + H_i)} \tag{8}$$

Based on Fresnel's equation, the reflectance of the interface is proportional to the difference of refractive indices. The differences in refractive indices of the air-snow and the snow-ice interface are relatively small compared with those of the ice-water interface, i.e., $R_1$ ($I_1$) and $R_2$ ($I_2$) are relatively small compared to $R_3$ ($I_3$). Given that $I_1$ and $I_2$ are small and are not related to the ice growth, we use a constant to represent them in the model. The backscattering coefficients should be proportional to the backscattered intensity $I_b$. Therefore, we suggest using Equation (9) to relate backscattering coefficients with the snow and ice thickness:

$$\sigma_0 = A + B \times e^{-K(H_s + H_i)} \tag{9}$$

where $\sigma_0$ is the backscattering coefficient, $A$, $B$, and $K$ are model parameters to be calibrated. The following strategy can be used to determine the parameters in an efficient way. Parameter $A$ generally ranges from 0 to 20 dB and is not very sensitive. Therefore, discrete values can be assigned to $A$ directly, such as 0, 1, …, 20. Then for each assigned parameter $A$, transforming Equation (9) into Equation (10) results in the logarithmic regression model:

$$(H_s + H_i) = -\frac{1}{K} \times \ln(\sigma_0 - A) + C, \ C = \frac{\ln(B)}{K} \tag{10}$$

where parameters $K$ and $C$ in Equation (10) can be determined using linear regression. The residual sum of squares for each set of $A$, $K$, and $C$ can be calculated and the parameter group with the lowest residual sum of squares was selected as the final estimates. The calibrated parameters are generally satisfactory as shown in Fig. 3 (b). It is possible that the regression model

yields negative LIT at the beginning of the ice seasons because the initial backscattering coefficient exceeds the range of data used in the regression. If that is the case, Equation (9) can be adjusted as Equation (11) to ensure that the initial LIT is non-negative, where $\sigma_{max}$ is the maximum backscattering coefficient during that ice season.

$$\sigma_0 = A + \sigma_{max} \times e^{-K(H_s + H_i)} \qquad (11)$$

### 3.3 Water level estimation for ice-covered lakes

Yang et al. (2021) developed a straightforward method to retrieve water levels for ice-covered lakes using T/P ad Jason-1/2/3 data. The basic concept of their method is to extract the first subwaveform from the double-peak waveform, and apply the 0.1 threshold retracking algorithm to the first subwaveform. By comparison with in situ water levels, lake ice thickness, and snow depth, Yang et al. (2021) suggested that the first subwaveform retracked with a 0.1 threshold (e.g., the red circle in Fig. 3(a)) is associated with the snow-ice interface and can be used as a good approximation to the free water surface. We noticed that

in the dual-threshold retracking algorithm (Li et al., 2022a), the first tracking gate (e.g., the first red triangle in Fig. 3(a)) is very close to the 0.5 threshold tracking point of the first subwaveform. By comparing the altimetric LIT with in situ LIT and snow depth, we found that the altimetric LIT is close to the total thickness of ice and snow for most cases, meaning that the first tracking gate is likely associated with the snow surface. Consequently, it remains a pending question whether the first subwaveform represents signals from the snow surface or the snow-ice interface.

In addition, for a given waveform as shown in Fig. 3 (a), the 0.1 threshold tracking gate (red circle) should be ahead of the 0.5 tracking point (the first red triangle), meaning that the associated surface height of the 0.1 threshold should be higher than that of the 0.5 threshold. But based on the mentioned two studies, the 0.1 threshold is related to the snow-ice interface while the 0.5 threshold is related to the snow surface. The inconsistency between the two studies causes ambiguity in determining the interface associated with the first subwaveform, thereby reducing the reliability of altimetric LIT and LSH for ice-covered

lakes.

LSHs for ice-covered lakes were first retrieved using different thresholds. The 0.1 threshold yields higher LSH than the 0.5 threshold for each waveform, meaning that a systematic bias exists between LSH time series from the 0.1 threshold and from the 0.5 threshold. To remove the systematic biases, we selected LSH retrievals during open water periods as the baseline, because observations obtained during open water periods are more stable and robust. For the open water period, the classic 0.1

and 0.5 threshold methods were directly applied to the waveform separately. During the ice-covered period, the 0.1 and 0.5 threshold methods were applied to the first subwaveform. All LSH retrievals for both ice-covered and open water periods derived with the 0.1 threshold were aggregated into one time series (*LSH_01*), and all LSH retrievals based on the 0.5 threshold were aggregated into the other (*LSH_05*).

Subsequently, the systematic biases (*bias*) between the two time series were calculated as the mean difference between *LSH_01*

and *LSH_05* during open water periods. Then we combined the two time series by concatenation of observations from *LSH_05*

during open water periods and those from (*LSH_01-bias*) during ice-covered periods, yielding the merged LSH time series for the entire study period. In Sect. 4.3, we show that the merged time series outperformed both *LSH_01* and *LSH_05*.

## 4 Results

### 4.1 Performance of the logarithmic regression model

To evaluate the performance of the logarithmic model we proposed (equation (10)) to convert backscattering coefficients into LIT, we compared it with the power function model used by Zakharova et al. (2021). As we mentioned in Sect. 3.2, our method can use waveform-based LIT to calibrate parameters in the logarithmic model and does not rely on in situ LIT. However, to evaluate the feasibility and potential of both models, we directly used in situ LIT in Baker Lake instead of waveform-based LIT to generate model parameters, which could represent the best performance of both models ideally. In addition, different

from Zakharova et al. (2021) who split in situ data into training and validation periods, we used all available data in each ice-covered season to calibrate parameters for both models due to the large variability of optimal parameter sets in different ice-covered seasons.

As illustrated in Sect. 3.2, the power function model assumes the LIT to be a power function of the accumulated backscattering difference. The LIT naturally starts from zero on the freeze-up date detected by Jason-1/2/3, because the accumulated

backscattering difference is zero at the beginning. It has been shown that the power function model is effective in estimating river ice thickness thinner than 1 m. But it is not clear if it is suitable for thicker ice conditions, e.g., in Baker Lake.

The maximum LIT in Baker Lake exceeds 2 m and for most time of the frozen season the LIT is over 1 m, which means that the LIT would increase rapidly at the beginning of ice seasons. Therefore, when Jason-1/2/3 first detects the lake ice in their 10-day revisit cycles, the LIT is not zero but could be several decimetres, which is not fully considered in the power function

model. The logarithmic model, however, is compatible with such kind of initial LIT conditions, as shown by Fig. 6. It is obvious that for the power function model, the initial LIT is estimated as zero but in situ measurements could range from 0.2 to 1 m (Fig. 6(a)).

The logarithmic model could well represent the initial LIT and its overall performance ($R^2$ of 0.90 and RMSE of 17 cm) is better than that of the power function model ($R^2$ of 0.77 and RMSE of 25 cm) even if the initial LIT data pairs are removed

from the power function model ($R^2$ of 0.78 and RMSE of 22 cm). In addition, underestimation of LIT is more severe in the power function model when the LIT exceeds 1.5 m, suggesting some saturation effects. Therefore, we suggest to use logarithmic models when the LIT exceeds 1 m or the LIT increases rapidly at the beginning of ice seasons.

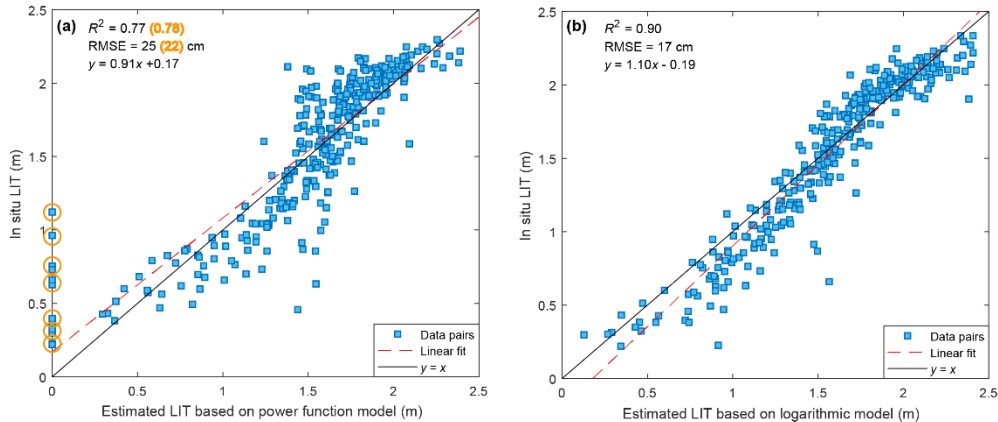

**Figure 6: Comparison between in situ and estimated LIT of Baker Lake based on (a) a power function model (Zakharova et al., 2021) and (b) a logarithmic model (this study). For both models, a separate set of parameters were derived for each ice season to best fit the in situ LIT. (a) and (b) are scatterplots of all matched data pairs from 2003 to 2019. Numbers in brackets denote metrics after the removal of outliers marked by yellow circles.**

## 4.2 LIT based on the combination of waveforms and backscattering coefficients

The accuracy of waveform-based LIT has been reported to be 0.15–0.2 m based on a comparison against in situ data (Li et al., 2022a). The waveform-based LIT data set used in this study is available online (https://doi.org/10.5281/zenodo.5528542). Here we mainly validated backscattering coefficient-based LITs against in situ thickness of lake ice and snow (Fig. 7). The overall performance of the backscattering coefficient-based LIT is close to that based on waveforms as summarized in the Table 3. Correlation coefficients (CCs) and RMSEs for Baker Lake, GSL, and Peipus Lake are 0.94, 0.80, and 0.76, and 24 cm, 17 cm, and 11 cm, respectively. As suggested by Zakharova et al. (2021), such accuracy is applicable in climate studies but may not meet the need for engineering purposes (e.g., ice roads).

The backscattering coefficient-based LITs using our method show metrics slightly lower than that of Zakharova et al. (2021) (RMSE: 7–18 cm). However, the relative errors between the two studies are similar, because the ice thickness in the previous study (Zakharova et al., 2021) is generally smaller than 0.8 m, while the LIT and snow thickness on GSL and Baker Lake could be over 1.5 m and 2 m, respectively. On the other hand, our method does not depend on in situ data and can be applied to ungauged lakes without in situ LIT measurements but with altimetric data. As we illustrated in Sect. 3.2, parameters used to convert backscattering coefficients into LIT can be calibrated against waveform-based LIT. The backscattered LIT shown in Fig. 7 and Fig. 8 was generated solely based on altimetry information. Calibration parameters and metrics for each lake and each ice season are available in supplementary materials (Table S1).

445 The backscattered LIT has advantages in estimating thin ice, albeit based on parameters calibrated against waveform-based LIT. In Peipus Lake, where the total thickness of snow and ice does not exceed 0.8 m, the waveform-based method can only retrieve the LIT at the very late phase of ice accumulation, resulting in limited observations each year. However, based on the limited waveform-based LIT, e.g., four or five observations over 0.5 m, the backscattered LIT can be generated to provide more complete tracking of the lake ice thickness, as shown in Fig. 7(e). The performance in Lake Peipus is relatively lower in 450 terms of CC, which is likely due to a higher contribution of signals from the lake surface snow, as the snow generally comprises 20–50% of the total thickness.

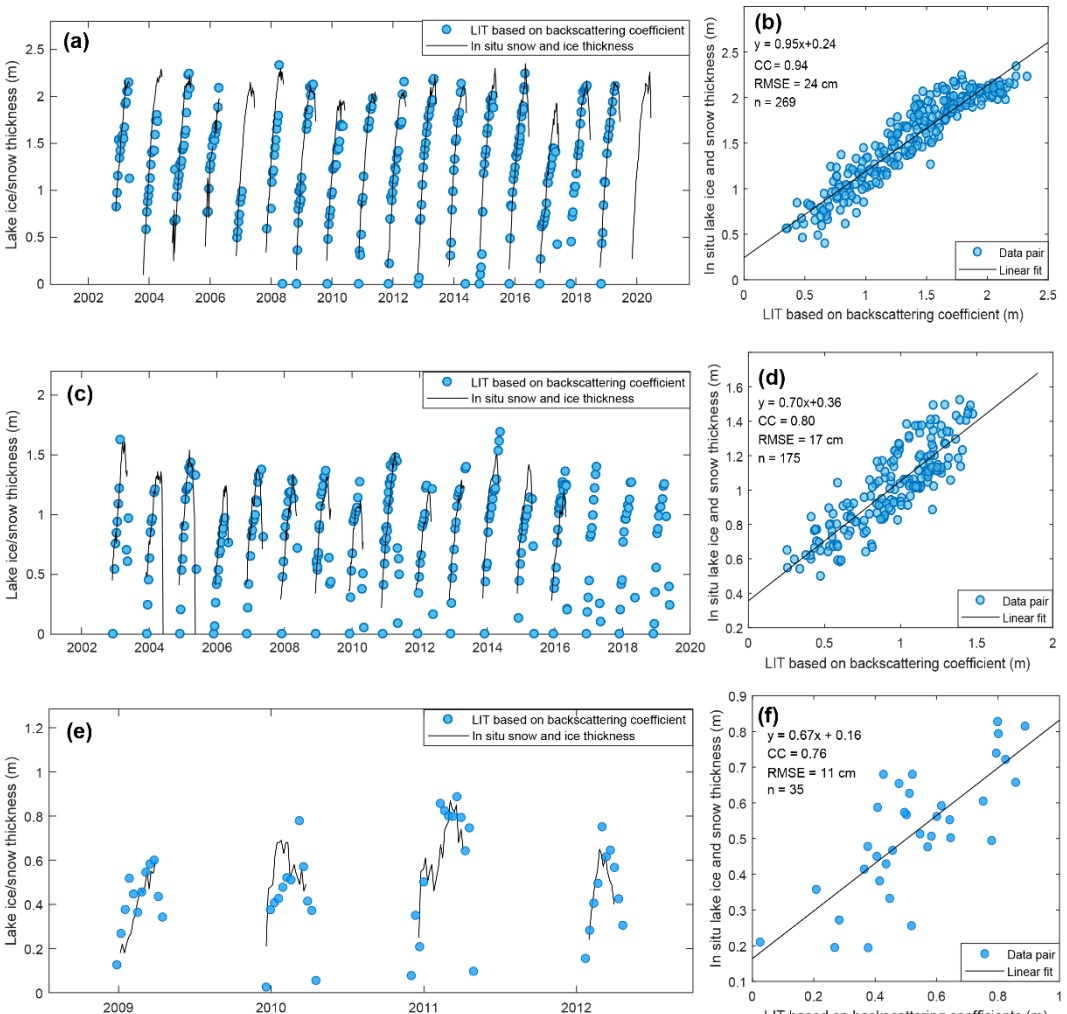

**Figure 7: Validation of backscattering coefficient-based LIT against total thickness of ice and snow. (a), (c), and (e) are time series for the backscattering coefficients-based LIT and the in situ lake ice and snow thickness in Baker Lake, GSL, and Peipus Lake. (b),**
455 **(d), and (f) are scatterplots of backscattering coefficients-based LITs and in situ lake ice and snow thickness in Baker Lake, GSL, and Peipus Lake, respectively.**

LITs in GSL and Baker lake are relatively large and the ice thickness grows rapidly at the beginning of the ice season, making these lakes not suitable for validating thin ice estimates. For Peipus Lake, the LIT is so small that for many years there is no available waveform-based LIT to calibrate parameters for the logarithmic model. To further assess the capability of backscattered LIT in the detection of thinner ice, we compared the altimetry-based LIT with modelled LIT in another two lakes with ice thickness of ~ 1 m, i.e., Hulun (117.38 °E 48.97 °N) and Har (93.21 °E 48.05 °N). Given different advantages of waveform-based LIT and backscattered LIT, we used an empirical method to merge the LIT time series based on waveforms and backscattering coefficients: for waveform-based LIT measurements, we reserved those larger than 0.7 m, and for backscattering coefficients-based measurements, we reserved those smaller than 0.7 m.

Another reason for selecting the two lakes above is that there is little snowfall in these lakes during ice-covered seasons (as shown in Fig. 8), which can reduce the impact of surface snow because the physical process of surface snow is complicated and could cause large uncertainty in modelled results (Han et al., 2019; Han et al., 2021). The waveform-based LIT in Hulun Lake was not sufficient to build a regression model (see Sect. 3.2) for each winter before 2014, so we only made cross-validation through 2014–2018. In Har Lake, the cross-validation was made through 2003–2018, as shown in Fig. 8. Overall, the merged altimetric LIT and model results agree well with each other in terms of an $R^2$ of 0.88 for Hulun Lake and 0.79 for Har Lake. But there is a relatively larger discrepancy in Har Lake, which is likely caused by the narrower cross-section of Har Lake and fewer available altimetric footprints.

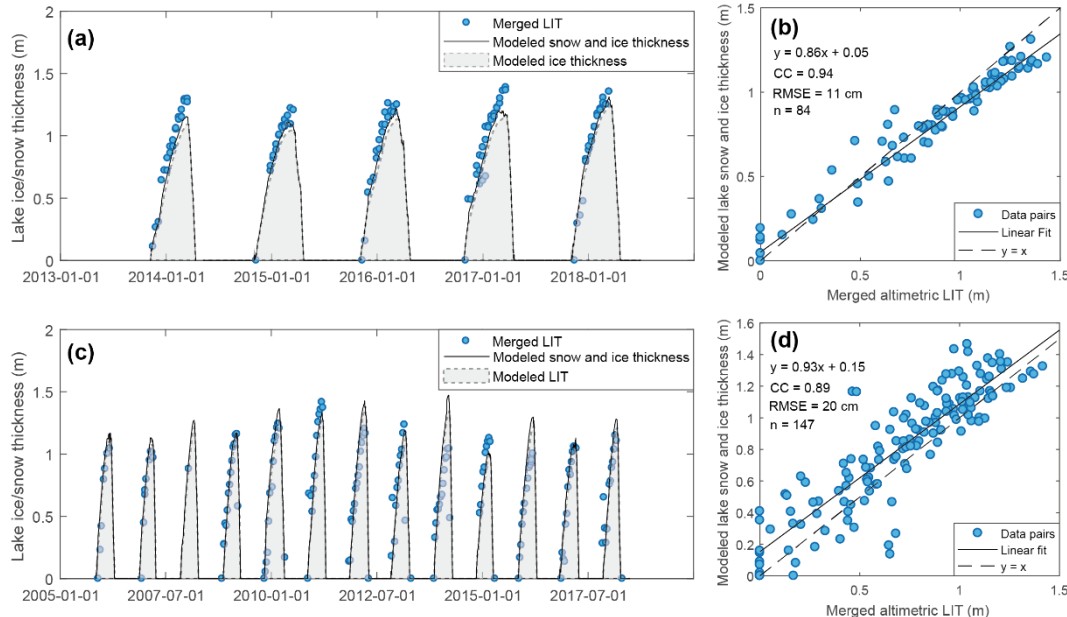

**Figure 8: Cross-validation between the merged altimetric LIT (waveforms and backscattering coefficients-based) and modeled lake snow and ice thickness. (a) and (c) are time series for merged altimetric LIT and modeled lake snow and ice for Hulun Lake and**

Har Lake, where blue dots denote merged altimetric LIT, shading areas denote modeled LIT, and black curves denote modeled lake ice and snow thickness. (b) and (d) are scatterplots for altimetric LIT and modeled lake ice and snow thickness for Hulun Lake and Har Lake, respectively.


**Table 3 Validation/Cross-validation metrics of altimetry-based LIT in five study lakes**

| Lake name | CC | RMSE (cm) | Reference data |
|-----------|-----|-----------|----------------|
| Baker Lake | 0.94 | 24 | In situ |
| Great Slave Lake | 0.80 | 17 | In situ |
| Peipus Lake | 0.76 | 11 | In situ |
| Hulun Lake | 0.94 | 11 | Modeled |
| Har Lake | 0.89 | 20 | Modeled |

### 4.3 Water level estimation for ice-covered lakes

The merged LSH time series were obtained by combining the LSH based on the 0.1 threshold and the 0.5 threshold method as
illustrated in Sect. 3.3. Results in Figure 9 show that with the systematic bias during the open water period removed, the LSH during ice seasons derived from the 0.5 threshold is higher than that derived from the 0.1 threshold, which contradicts the intuition that the 0.5-threshold-based LSH should be lower than the 0.1-threshold-based LSH. It suggests that different choices of thresholds would result in different backscattering interfaces when the lake surface is covered with snow and ice. For instance, Yang et al. (2021) suggested that the 0.1 threshold corresponds to the snow-ice interface while Li et al. (2022a) using
a 0.5 threshold suggested that their results were close to the air-snow interface. We further discuss causes of this phenomenon in Sect. 5.1, indicating that conclusions from Yang et al. (2021) and Li et al. (2022a) are not contradictory.

When comparing LSH derived from the 0.1 threshold and the 0.5 threshold, we noticed that the 0.5 threshold-based LSH retrievals have a more robust performance during the open water period (as shown in supplementary Fig. S2), corroborated by previous studies (Davis, 1997). For the ice-covered period, the 0.1 threshold-based LSH retrievals are very close to the
hydrostatic water level as suggested by Yang et al. (2021). Therefore, we merged the two LSH time series (after removing their systematic bias during open water periods) by reserving the 0.5 threshold-based LSH during the open water period and 0.1 threshold-based LSH for the ice-covered period to improve the overall performance of water level estimation. The improvements of the merged LSH time series compared to those based on a single threshold are shown in Table 4.

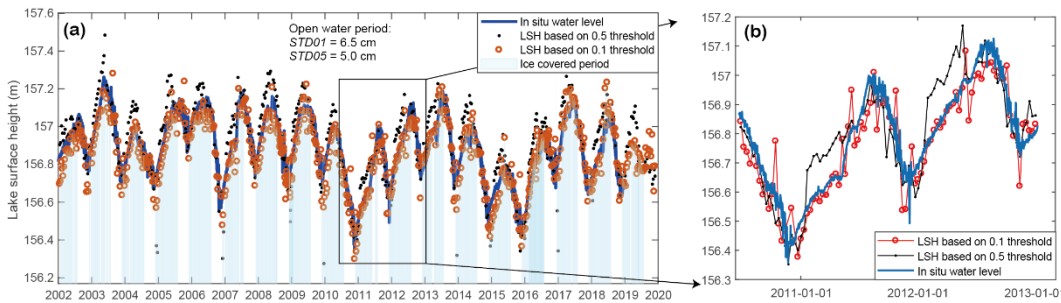

Figure 9: Comparison between LSH estimates using different thresholds in GSL. (a) shows time series for altimetric LSH based on different thresholds and in situ water levels. (b) is an enlarged view for the LSH time series during 2011–2013. The blue curve denotes in situ water levels, red dots denote LSH based on a 0.1 threshold, black dots denote LSH based on a 0.5 threshold, and light blue shading areas denote ice-covered seasons.

Table 4 Improvements of merged LSH compared with the LSH based on a single threshold validated against in situ water levels

| Lake name | Merged LSH RMSE (cm) | 0.1-threshold RMSE (cm) | 0.5-threshold RMSE (cm) |
|---|---|---|---|
| GSL | 7.1 | 8.0 | 14 |
| GBL | 8.1 | 9.4 | 10.6 |
| Athabasca Lake | 9.8 | 11.5 | 12.5 |
| Winnipeg Lake | 10.2 | 11.7 | 19.4 |

We derived altimetric LSH time series for four lakes: Athabasca Lake, GBL, GSL, and Winnipeg Lake. Results were validated with in situ water levels using RMSEs (Fig. 10). As mentioned in Sect. 3.3, the LSH time series based on both 0.1 and 0.5 thresholds were derived first and then merged into one time series by removing the systematic bias during open water periods. Compared to in situ measurements, the 0.1 threshold-based LSH retrievals outperformed the 0.5 threshold-based LSH retrievals in representing hydrostatic water levels during ice-covered periods, while the 0.5 threshold-based LSH better represents in situ water levels during open water periods (Fig. 6). Therefore, the merged LSH time series should outperform both 0.1 and 0.5 threshold-based LSH time series. We did notice an improvement of 1.5–2 cm in RMSE for each lake as shown in Table 4. Overall, the metrics of the derived water levels are consistent with those from (Yang et al., 2021) in GSL, GBL, and Athabasca Lake. However, a direct comparison with metrics from (Yang et al., 2021) would be inappropriate, as we used different ground tracks and different gauging stations.

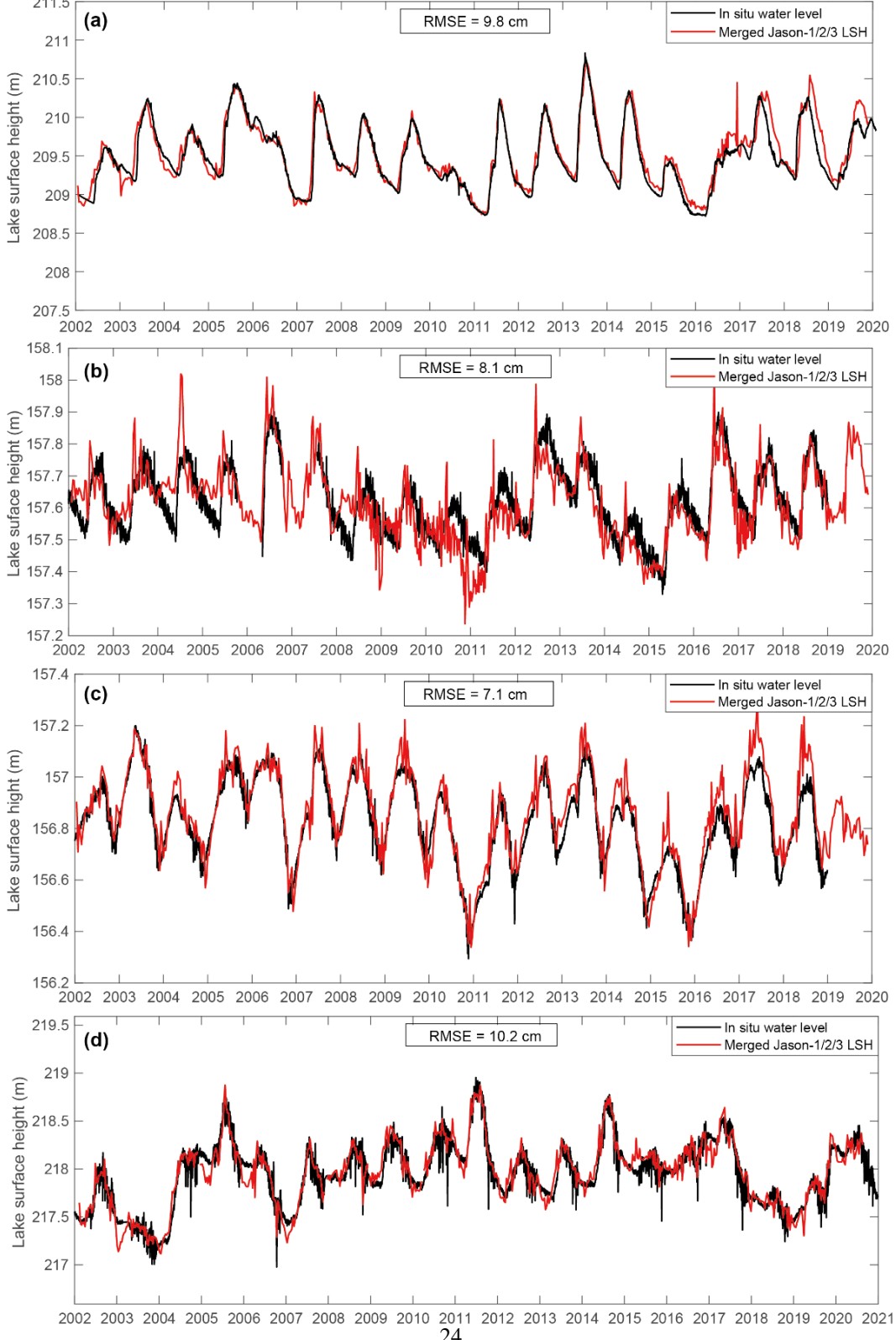

**Figure 10: Time series of merged altimetric water levels in Athabasca Lake, GBL, GSL, and Winnipeg Lake. Black curves denote in situ lake water levels and red curves denote merged altimetric LSH for (a) Athabasca Lake, (b) GBL, (c) GSL, and (d) Winnipeg**
**Lake. Note that there are systematic biases between Jason-1, Jason-2, and Jason-3 data, which were removed by comparing the mean LSH during the overlapping periods between Jason-1 and Jason-2 (Jul 2008–Jan 2009), and Jason-2 and Jason-3 (Feb 2016– Sep 2016).**

## 5 Discussion

### 5.1 Conceptual explanation on differences in LSH derived from different thresholds

525    Normally a higher threshold of waveform retracking yields a lower LSH, but comparison shown in Fig. 9 indicates that LSH based on the 0.5 threshold is higher than that based on the 0.1 threshold when the systematic bias during open water periods is removed. Here we provide a conceptual explanation as to why such a phenomenon occurs. As shown in Fig. 11 (a–c), the pulse-limited satellite altimetry sends a microwave pulse with a certain width to the open water surface (with a calm wave surface) and the illuminated area gradually increases to the maximum when the upper bound of the pulse reaches the water

530    surface. Ideally, the largest illuminated area is associated with the peak of the radar waveform (Fig 11 (c)). For threshold retracking methods, a portion of the maximum power is used to mark the time when the radar pulse reaches the backscattering surface. For instance, the 0.1 threshold method essentially means that the moment when the echoed radar pulse surpasses a 10% of the waveform peak is selected to be the time when the radar pulse reaches the lake surface (Fig 11 (a)).

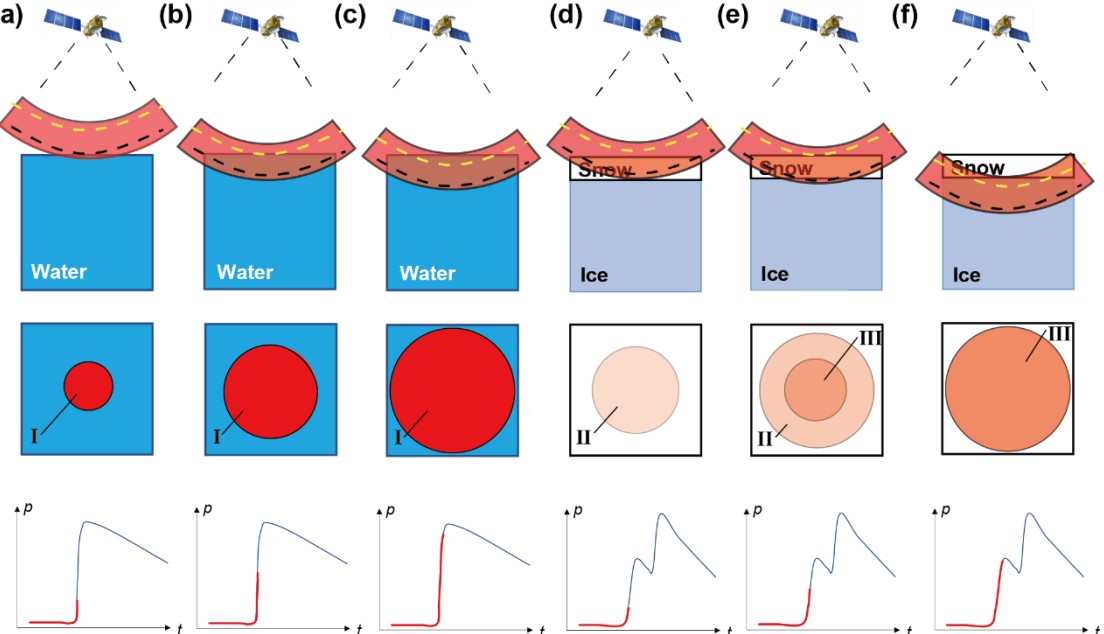

535    **Figure 11: A schematic diagram of pulse-limited radar footprints/illuminated areas and associated waveforms on open water (a–c) and ice-covered (d–f) lakes. The first horizontal panel shows the sideview of the radar pulse and backscattering interfaces. Black**

and yellow dashed curves denote hypothetic spheres within the radar pulse associated with the 0.1 and 0.5 thresholds, respectively. The second horizontal panel represents the illuminated areas in a vertical view, where circular area I denotes backscattering from the water surface (a–c), circular area II denotes backscattering from snow layers (d and e), circular area III denotes surface backscattering from the snow-ice interface (e and f). The third horizontal panel shows waveforms associated with illuminated areas in the second panel. *P* represents the returned power and *t* represents the time/gate. The red curve indicates the part of the waveform that has emerged, whereas the blue curve indicates the rest part. Waveforms in (a–c) indicate moments when a 10%, 50%, and 100% of the waveform peak is met, respectively. Waveforms in (d–f) indicate moments when a 10%, 50%, and 100% of the peak in the first subwaveform is met, respectively.

To make the process of the threshold retracking more visible, we assume a sphere within the pulse as shown by the black dashed curve in Fig. 11. The hypothetical sphere is assumed to have a certain distance/time lag from the lower bound of the radar pulse and we name it the 0.1-shpere for simplicity. The time when the 0.1-sphere reaches the lake surface indicates that the 0.1 threshold is met and an LSH is recorded. The recorded LSH is the absolute height of the radar pulse with respect to the reference ellipsoid or geoid (e.g., for Fig. 11 (a–c), 100 m, 99.5 m, and 99 m). Similarly, we assume a sphere for the 0.5 threshold method and name it the 0.5-sphere, as shown by the yellow dashed curve in Fig. 11. For open water periods, received waveforms only come from the air-water interface and the time lag (range difference) between the 0.1-spere and the 0.5-sphere is a relatively stable value with some fluctuations caused by varying wave heights.

When the lake is covered by snow and ice, the illuminated areas become more complicated (Fig. 11 (d–f)). The waveform consists of information from multiple backscattering surfaces and volume backscattering. Consequently, there could be multiple peaks in the waveform. Here, we only focus on the first peak because it is most relevant to either the snow surface or snow-ice interface. Based on previous studies (Atwood et al., 2015; Beckers et al., 2017), we assume the first waveform peak to occur when the upper bound of the pulse reaches the snow-ice interface, as shown in Fig. 11 (f). Then we apply the 0.1 and 0.5 threshold methods to the first peak (Fig. 11 (d–e)). If there is no snow cover and volume scattering, the 0.1 threshold would be met when the 0.1-sphere reaches the ice surface and the 0.5 threshold would be met when the 0.5-sphere reaches the ice surface. However, volume scattering from snow layers contributes to the returned power so that the backscattered energy increases faster than the case without volume scattering. Consequently, the time lag (range difference) between the 0.1-sphere and the 0.5-shpere is compressed. Meanwhile, the LSH recorded in Fig. 11 (d–f) could be 99.6, 99.4, and 98.9 m. The associated systematic bias between the 0.1-threshold (Fig. 11 (d)) and the 0.5-threshold (Fig. 11(f)) becomes smaller than the case of open water (Fig.11(a–b)). Therefore, with the systematic bias during the open water period removed, the 0.5-threshold-based LSH would be larger than the 0.1-threshold-based LSH in ice covered seasons.

Here we used the LSH during 2008–2009 in GSL as an example to illustrate the case above. Fig.12 shows the original LSH based on the 0.5-threshold method and the 0.1-threshold method with systematic bias. During the open water period, the average difference in LSH between the 0.1-threshold and the 0.5-threshold is 0.46 ± 0.04 m, while that during the ice-covered

season decreases to 0.36 ± 0.02 m as shown in Fig. 12. With the system bias during the open water period removed, the LSH

based on 0.5-threshold will inevitably exceed that based on the 0.1-threhold during the ice-covered period.

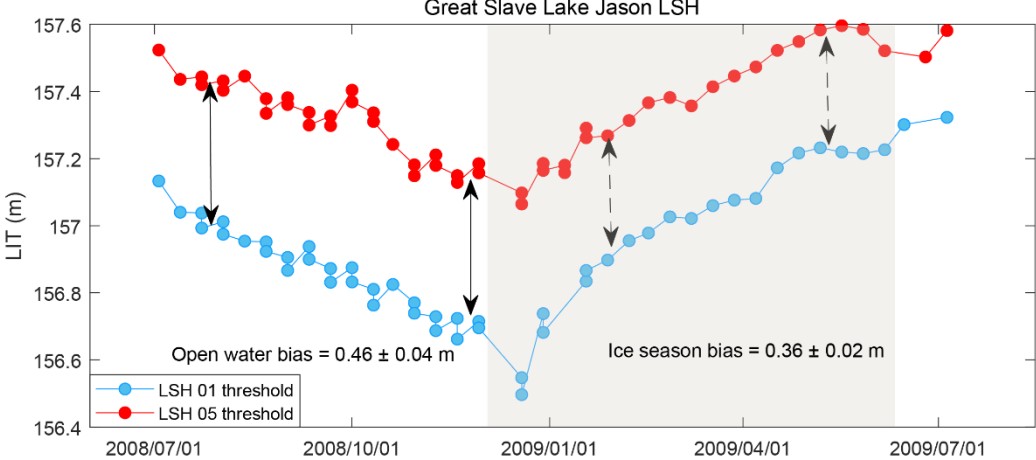

**Figure 12: Different systematic biases between the LSH based on the 0.1 threshold and the 0.5 threshold in different seasons. Red and blue curves denote original LSH based on the 0.1 threshold and the 0.5 threshold in GSL during 2008—2009. The gray shade**
**represents the ice-covered season.**

### 5.2 Uncertainty and limitations

The main source of uncertainty in LIT estimation is lake-surface snow cover. As a result of the impact of snow cover, the accuracy of remotely-sensed LIT is in general 0.1–0.2 m in current studies. As discussed in Sect. 5.1, lake surface snow could influence radar waveforms as well as backscattering coefficients. In addition, some physical variables or processes related to
snow and ice not considered in our model could also contribute to the uncertainty of the results.

Regarding the backscattered LIT, we did not consider the effect of volume scattering. Volume scattering is caused by snow particles and air bubbles captured inside ice, while ice-bottom scattering is controlled mostly by the roughness and dielectric constant ($\varepsilon$) of the ice/water interface. For dry snow, volume scattering from snow cover can increase backscattering coefficients of Ku-band radar obtained from frozen lakes (Gunn et al., 2015). Based on Kim et al. (1984), thicker snow cover
contributes more to backscattering coefficients due to enhanced volume scattering. Consequently, given the same ice condition, backscattering coefficients obtained from thick snow-covered lakes should be larger, which could result in underestimation of the LIT. On the other hand, wet snow can hardly be penetrated by microwave and could largely reduce the backscattered energy, resulting in overestimation of the LIT.

For the waveform-based LIT, the most important physical property is the $\varepsilon$ of snow and ice, as it determines the speed of light
within snow and ice and the timing of reflected signals from different interfaces (higher $\varepsilon$ corresponds to the lower speed of light). During the ice accumulation process, the $\varepsilon$ of ice is relatively stable. The $\varepsilon$ of dry snow is almost solely dependent on

snow density (Tiuri et al., 1984), which can be approximated with $\varepsilon = 1 + 2\rho$, where $\rho$ is the relative snow density (with respect to water). However, we used the same constant $\varepsilon$ for both ice and snow, which is a compromise as we do not have any prior information related to snow depth and density. Because the waveform-based method measures the time difference between

different interfaces, at the beginning of ice and snow accumulation, our method could slightly underestimate the total thickness of snow and ice because snow has a smaller $\varepsilon$ and a larger speed of light. As the snow becomes denser during the frozen period and the speed of light becomes slower in snow, the waveform-based LIT could be closer to the total thickness of snow and ice.

As for the uncertainty of water levels, apart from the impact of snow and altimeter range resolution, the source of uncertainty is associated with remaining systematic biases. Water level time series for ice-covered lakes are based on the connection of

observations from Jason-1, Jason-2, and Jason-3 after the removal of systematic biases. To identify systematic biases, mean water levels from different sensors during overlapping periods are compared, which is a technique commonly used. However, it is not clear to what extent the remaining systematic biases contribute to the uncertainty in the entire water level series. We estimated that the upper limit of the remaining systematic biases is ~ 5 cm. A detailed description of uncertainty quantification can be found in the Supplementary Information (Supplementary Text 1).

**5.3 Implications for future studies**

Based on the discussion in Sect. 5.1, different thresholds correspond to different interfaces (e.g., air-snow and snow-ice interfaces) in ice-covered seasons. If the estimation of LSH with different threshold methods can be further improved, it is possible to discriminate the snow depth from the altimetric LIT. The relationship between backscattering coefficients and the surface snow depth can be further investigated, which could facilitate more robust modelling of lake ice and snow based on

backscattering coefficients. It could also facilitate more sophisticated validation of lake ice models containing snow processes.

The method developed here has the potential to be used in early satellite altimetry missions including T/P, ERS-1/2, as well as some follow-on missions such as Jason-CS (Scharroo et al., 2016), extending remotely sensed LIT to three decades and wider spatial coverage. However, it should be investigated whether the developed method is suited for Ka-band altimeters or SAR altimeters such as SARAL/AltiKa, CryoSat-2, and Sentinel-3. This is because the penetration ability of Ka-band microwave

and Ku-band microwave in ice and snow could be quite different and the pulse-doppler-limited waveforms (or beam-limited waveforms, e.g., CryoSat-2) are different from pulse-limited waveforms.

**6 Conclusion**

This study presents an effective method to retrieve LIT based solely on altimetric data (including waveforms and backscattering coefficients), which is applicable to lakes without in situ LIT measurements. We also investigate water level estimation for

ice-covered lakes by merging LSH time series derived from different threshold retracking algorithms. Major findings are as follows:

(1) A logarithmic regression model could be more effective in converting backscattering coefficients into LITs than a previously used power function model, in terms of an $R^2$ of 0.90 and an RMSE of 17 cm for the developed logarithmic model.

(2) Validated against in situ measurements and modelled lake ice and snow thickness, the developed altimetric LIT estimation method combines the advantages from the waveform-based method (physically-based, sensitive to thick ice) and the backscattering coefficient-based method (sensitive to thin ice). The accuracy (or RMSE) of the merged altimetric LIT is ~ 0.2 m for the study lakes.

(3) Merging LSH time series derived from different threshold retracking algorithms (0.1 and 0.5 thresholds) can improve the
performance of water level estimation for the entire study period by 1.5–2 cm, compared to the estimation with single threshold methods in terms of RMSE among the study lakes.

(4) Different threshold retracking algorithms (0.1 and 0.5 thresholds) can represent different backscattering surfaces for ice-covered lakes. Compared to the same baseline (LSH during open water period), the 0.1 threshold could represent the snow-ice interface while the 0.5 threshold could be closer to the air-snow interface.

Overall, we provide a more robust and adaptive method for remote sensing of LIT and LSH for ice-covered lakes without in situ observations. The differential impact of lake surface snow on different threshold methods and its implications in future research related to altimetric LIT and water level estimation are discussed. This study facilitates a better interpretation of satellite altimetry signals from ice-covered lakes and provides opportunities for a wider application of altimetry data to the cryosphere.

**Author Contributions:**

Conceptualization, X.L. and D.L.; methodology, X.L. and D.L.; data curation, X.L.; writing-original draft preparation, X.L.; writing-review and editing, D.L., Y.C., T.L., J.L., M.A.H., and M.M.M.

**Conflicts of Interest:** The authors declare no conflict of interest.

**Data and Code Availability Statement:**

The data and code presented in this study are available on a reasonable request from the corresponding author.

**Acknowledgements**

This work was supported by the Major Science and Technology Projects of Inner Mongolia Autonomous Region (2020ZD0009) and the National Natural Science Foundation of China (92047301). Reviewers and editors' comments that are useful to

improve this study and manuscript are highly appreciated. The authors sincerely thank the Canadian Ice Service for providing
in situ lake ice and snow thickness, the Water Survey of Canada for providing the in situ water levels, and the AVISO+ for
providing the satellite altimetry data that enabled us to conduct this study.

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
