# Peer review of "Ice thickness and water level estimation for ice-covered lakes with satellite altimetry waveforms and backscattering coefficients"

_The Cryosphere, 2022_

## Referee Comment (RC1)

**Ice thickness and water level estimation for ice-covered lakes with satellite altimetry waveforms and backscattering coefficients.**

by Xingdong Li, Di Long, Yanhong Cui, Tingxi Liu, Jing Lu, Mohamed A. Hamouda, and Mohamed M. Mohamed

The paper titled "Ice thickness and water level estimation for ice-covered lakes with satellite altimetry waveforms and backscattering coefficients" by lead author Xingdong Li and co-authors explored radar altimetry data to infer lake ice thickness. By conducting the study over 6 lakes, the authors improved existing methods of ice thickness estimation using the radar altimetry and reported improved accuracy of their retrievals by comparing altimetry-derived ice thickness estimation and those from the in situ gauge records and from the modelling. The authors have done an considerable efforts of describing the details and nuances of the radar signals scattering within the ice and how it matches the assumptions they applied in the modified methods.

My major concern with the paper is the lack of clarity and consistency in presentation of the method and results. While the theory of the methods were well described, the implementation routine sometimes is unclear.

The supplementary material provides an excessive theoretical information and can be reduced to several lines describing how exactly the inter-satellite bias was calculated (along-track point-by-point approach or on the cycle basis, the length of the tandem mission phase for each tandem, and the obtained bias for each tandem - average or median bias plus the standard deviation or the range. This information is crucial for evaluation of the results. The authors claims several times in the texts that the presented method does not take the in situ observations, while in the Methods section the calibration of several parameters of equations was mentioned. More information on this calibration is required (period of calibration, values of calibrated parameters, uncertainties).

Please see below my comments. I would recommend the authors provide less theory and more implementation techniques in the Methods section; better prove the findings providing consistent (the same) set of statistics across the text in the tables for each lake; reduce the Supplementary Materials to several lines and introduce them into the main text.

Overall, I think the paper needs a major revision.

**General comments.**

For lake surface height retrieving the authors tested two thresholds and removed the systematic bias between the heights obtained using these thresholds and calculated for the open water period. Then, finally, they found that the winter water heights obtained with 0.1 threshold are lower than the heights obtained with the 0.5 threshold (fig.6), which contradicts with the theory well illustrated in the Figure 3a. The explications provided in the Discussion section 5.1 are quite interesting. However, to be convinced I would like to see an example of quantitative evaluation for one case: compressed pulse length/gate width/LIT/snow depth/ values of open water bias etc.

I would expect that the summer height bias due to the thresholds applied is the value variable from cycle to cycle as it depends on the leading edge width, which is the proxy of the surface roughness (wave height). The leading edge width (LEW) of a specular waveform sampled during calm water conditions may be similar to the LEW of a waveform over the 20-40 cm ice with some snow cover.  The open-water bias should be deduced from these open-water specular waveforms. Otherwise, the solid proofs are necessary (seasonal plots, or the bias value for waveforms of different peakiness, for example). And again, the tables of estimates with sets of descriptive statistics will help to follow and understand the authors' logic.

The subsection Uncertainties and Limitations is only qualitative and somewhat naive: a short paragraph on what the authors expect from the snow effect on the waveform and consequently on the LIT retrievals will be beneficial for the subsection.

**Specific comments.**

Line 83. Why the approach developed in Becker et al., 2017 for SAR waveforms is not compatible with the conventional altimetry waveforms?. SAR waveform is more specular, however the approach remains the same: retracking of the sub-waveforms.

Line 107. The word "paradox" is not good here. "Discrepancy? "
Line 121. I did not find a method based on "combination of the backscatter and waveform". For me it was the combination of LIT retrieved from the waveform and from the backscatter.

Lines 121-123. I would say "improved" instead of "novel". "Novel" means really a new approach and not modification of function type or threshold value.

Section 2.1. Please, re-write the section providing uniform description of the environmental /climate conditions for each lake. I can't evaluate the severity of the climate comparing the mean annual temperatures with mean July temperatures. The mean (or range) of negative temperatures or T of most cold winter month will tell more in LIT studies.

Lines 140 -142. The sentence is not dedicated to the Study area description.

Figure 1. Please, unify the scale for the three small lakes (lower panel) 1 cm = 100 km. In next sections you argue that the Har Lake is small and it was difficult to obtain good estimates due to its size. However, it seems that the Jason - lake cross-section length is quite similar for the Baker and Hur lakes.

Line 147. only for the ocean topography. Ice sheets studies appeared later.
Lines 149. earlier studies of Becker et al , 2017 and Duguay et al.,2018 should be cited here as well.
Lines 160-162. SGDR Jason product contains as well the range retrieved with Ice (OCOG) retracker widely used in inland water studies. (see S. Calmant, F.Frappart; S.Biancamaria etc. articles).
Lines 169-170 . please, check the prepositions in the sentences. Move the information on location of LIT in situ stations into the Table 1. In the Table 1, provide consistent with LIT station coordinates (in decimals). In this lines it would be better to mention that the in situ LIT is observed near the coast, where the ice growth and snow-on-ice  conditions can differ from open area of the lakes (especially GSL). See different studies of C.Duguay team for details.

Table 1. Please summarise here all data used for validation, water level, LIT gauges, model simulation.
Line 185. Please, give here 2-3 lines about 1-D lake ice model developed by Li et al.,(2022). What does it mean "remote sensing" for this model? What is the accuracy of the LIT simulations for the lakes selected for this study ?

Line 217. Better to speak "delay-Doppler" or "SAR" altimetry when meaning SC2 or Sentinel-3. SAR altimetry can be seen as the beam-limited only in along-track direction.
Line 220. and in other places : replace wave for waveform.
Line 240. Check c and $c_i$ in formulations; provide the reference on c in the ice.
Line 253. Delete "quite"
Line 269. Delete "A possible reason". This is the main reason. Provide a reference.
Line 272. replace "increases(decreases)" on " is high(low)"

Line 280. Explain STD, variability in spatial domain?

Line 285. I am not agree. The Sig0 rise during this phase is due to water-on-ice appearance or decreasing of the penetration depth (volume scattering) caused by high water content of the melting snow, unless you prove your statement with the solid references.

Line 320. "$I_1$ does not change with LIT" is also only the assumption. If I understood $I_1$ is the surface echo plus snow volume echo. It can change during the winter due to the changes in the snow (densification, redistribution) and ice (thermal or mechanical deformation resulting to changes in ice surface roughness).

Equation 8. Explain, please, why in the Eq2 "-2kHi" appeared.

Equation 10. Insert space between equations

Line 361 replace the word "paradox"

Line 369. Please, rephrase the sentence in more scientific manner. Phrase "investigate LSH for ice-covered lakes" is scientific slang.

Line 374.  All LSH **retrievals**...

Lines 377-381 These lines describe the results. Move them to the Result section. See my general comment dedicated to the evaluation of the open-water altimetric range bias obtained with 0.1 and 0.5 thresholds.

Lines 383. What does it mean "more robust performance"? Please, illustrate with the statistics comparing with corresponding reference time series. Remind in the text what Davis (1997) investigated: inland waters, which retracker (OCOG ) ?.

Lines 385-387. Not clear how it was implemented:  by concatenation of [ $LSH0.5_{openwater}$ ; ($LSH0.1_{ice}$ - bias)] ?

Line 423. The subsection 4.1 was also based on altimetric measurements. Change the title of the subsection 4.2 for more specific.

Line 424. Please, provide here the uncertainties found in the cited study. Not clear why you did not compare your waveform-based retrievals with the in situ observations. Did you use exactly the same areas, same tracks, same codes for selection of the sub-waveforms as in the Li et al., 2022 or you used LIT provided by these authors ?

Lines 426-430. Summarise the statistics for each lake in a Table. Be consistent providing coefficients of correlation or determination.

Line 432. Why your method (backscatter-based in logarithmic approximation ???) does not depend on availability of in situ observations. If I understood from the lines 339-351 you calibrated the parameters $K$, $A$ and $C$ ?.

Section 4.2 In the Figure 3 only the equation for GSL was provided. Please, in the section 4.2 give the table with 1) the K, A, C parameters for each lake, 2) with the period used for calibration, 3) period used for validation, 4) accuracy of the LIT for validation period (RMSE, correlation coef.).

Lines 440-445. Some reasoning based on demonstrations via figures or tables of statistics is necessary to prove why waveform-retrieved LIT was used for thick ice, while the backscatter-based LIT retrievals were used for thin ice range.

Line 453. For me, the track length within lakes Baker and Har looks the same (Figure 1), so the reason given in the text is not strong.

Figure 9. For simulated LIT+SnowDepth provide the line as well. The gray shadow-black line difference is not visible.

Line 466. What the "effective water level" is? Hydrostatic water level or water-ice interface?

Line 468. improvement comparing to what?. I prefer to see RMSE for each lake compared to the RMSE of "what this improvement refers to". Please, give the metrics found in Yang. etal 2022.

Figure 10. and everywhere. STD and RMSE statistics should be round to centimetre. The accuracy of the altimetry height retrievals over the inland water objects is still from several centimetres - to several tens of centimetres.

Lines 480-485. For me, the variability of the RMSE and STD between the lakes is low. Moreover, the STD is almost equal to RMSE, so keep only the RMSE. The explication of observed inter-lake uncertainties provided in this paragraph is unrealistic. The text does not correspond to the figure.

For the Discussion Section see my general comments.

---

## Author Comment (AC1)

Response Letter

Reviewer Comments 1

The paper titled "Ice thickness and water level estimation for ice-covered lakes with satellite altimetry waveforms and backscattering coefficients" by lead author Xingdong Li and co-authors explored radar altimetry data to infer lake ice thickness. By conducting the study over 6 lakes, the authors improved existing methods of ice thickness estimation using the radar altimetry and reported improved accuracy of their retrievals by comparing altimetry-derived ice thickness estimation and those from the in situ gauge records and from the modelling. The authors have done an considerable efforts of describing the details and nuances of the radar signals scattering within the ice and how it matches the assumptions they applied in the modified methods.

My major concern with the paper is the lack of clarity and consistency in presentation of the method and results. While the theory of the methods were well described, the implementation routine sometimes is unclear.

The supplementary material provides an excessive theoretical information and can be reduced to several lines describing how exactly the inter-satellite bias was calculated (along-track point-by-point approach or on the cycle basis, the length of the tandem mission phase for each tandem, and the obtained bias for each tandem - average or median bias plus the standard deviation or the range. This information is crucial for evaluation of the results. The authors claims several times in the texts that the presented method does not take the in situ observations, while in the Methods section the calibration of several parameters of equations was mentioned. More information on this calibration is required (period of calibration, values of calibrated parameters, uncertainties).

Please see below my comments. I would recommend the authors provide less theory and more implementation techniques in the Methods section; better prove the findings providing consistent (the same) set of statistics across the text in the tables for each lake; reduce the Supplementary Materials to several lines and introduce them into the main text.

Overall, I think the paper needs a major revision.

Response: Thanks for all these insightful and constructive comments. We have improved the clarity and consistency of the method and the results. We also reduced supplementary materials following the reviewer's suggestions. As for the reviewer's concern about the use of in situ data in model calibration, we have revised related sections to clarify why our method does not rely on in situ data. In brief, the parameters in the proposed logarithmic model used to convert backscattering coefficients into LIT, can be calibrated against the waveform-based LIT as opposed to in-situ data, when no in situ measurements are available. We used in-situ LIT to calibrate the parameters only in Section 4.1, to test the feasibility of the proposed logarithmic model and to explore

the potential of the model under 'ideal conditions' (See also our response to Comment 35). The reviewer's comments are addressed point-by-point in the following.

**General comments**

For lake surface height retrieving the authors tested two thresholds and removed the systematic bias between the heights obtained using these thresholds and calculated for the open water period. Then, finally, they found that the winter water heights obtained with 0.1 threshold are lower than the heights obtained with the 0.5 threshold (fig.6), which contradicts with the theory well illustrated in the Figure 3a. The explications provided in the Discussion section 5.1 are quite interesting. However, to be convinced I would like to see an example of quantitative evaluation for one case: compressed pulse length/gate width/LIT/snow depth/ values of open water bias etc.

I would expect that the summer height bias due to the thresholds applied is the value variable from cycle to cycle as it depends on the leading edge width, which is the proxy of the surface roughness (wave height). The leading edge width (LEW) of a specular waveform sampled during calm water conditions may be similar to the LEW of a waveform over the 20-40 cm ice with some snow cover. The open-water bias should be deduced from these open-water specular waveforms. Otherwise, the solid proofs are necessary (seasonal plots, or the bias value for waveforms of different peakiness, for example). And again, the tables of estimates with sets of descriptive statistics will help to follow and understand the authors' logic.

The subsection Uncertainties and Limitations is only qualitative and somewhat naive: a short paragraph on what the authors expect from the snow effect on the waveform and consequently on the LIT retrievals will be beneficial for the subsection.

Response: The reviewer suggests: (1) providing a quantitative example for the discussion in Section 5.1; (2) deriving the open-water bias from specular waveforms; and (3) revising uncertainty analysis in Section 5.2 to provide a brief discussion about the snow effect on the waveform and LIT retrievals.

(1) For the quantitative example, here we use the lake surface height (LSH) of 2008–2009 in GSL as an example to quantify the discussion in Section 5.1. During the open water period, the average difference in LSH between the 0.1-threshold and the 0.5-threshold is 0.46 ± 0.04 m, while that during the ice-covered season is 0.36 ± 0.02 m as shown in Fig R1 (a) below. With the system bias during the open water period removed, the LSH based on 0.5-threshold will inevitably exceed that based on the 0.1-threhold during the ice-covered period.

In Fig R1 (b), we further compared LSH based on different thresholds (with bias removed) with in situ lake water level, air-snow interface, snow-ice interface, and ice-water interface. The height of the air-snow interface, snow-ice interface, and ice-water interface was calculated based on in situ LIT, snow depth, and an assumed snow density of 0.2 g/cm$^3$ and ice density of 0.917 g/cm$^3$. It is shown that LSH based on the 0.1-threshold is close to the snow-ice interface, while the 0.5 threshold

is overall higher than the snow-ice interface during the ice-covered period. Although, in this case, the 0.5-threshold-based LSH is not very close to the air-snow interface as we assumed in this study, the difference between the 0.5-threshold-based LSH and the 0.1-threshold-based LIT is still related to the snow and ice accumulation which could facilitate future studies to better resolve different interfaces for ice-covered lakes.

[Figure]

Fig. R1 A quantitative example of the winter LSH deviation obtained with different threshold methods. (a) Jason-measured lake surface height (LSH) series with different threshold methods and their systematic biases in GSL during 2008−2009. (b) Jason-measured LSH time series (with bias removed) and in situ measured LSH, air-snow interface, snow-ice interface, and ice-water interface in GSL during 2008−2009.

(2) To derive systematic bias from specular waveforms, we used Pulse Peakiness (PP) to discriminate specular waveforms from diffuse waveforms. PP is calculated as the peak wave power divided by the averaged wave power. Typical specular waveforms have a PP value over 10, as shown in Fig R2(a). However, we noticed that for large lakes in this study, specular waveforms mainly appear during the freezing or melting stage when the lake is covered by thin ice (Fig R2(c)). Even though some high PP values were detected during the open water period, the associated waveforms were

mostly contaminated invalid waveforms that we have already removed from the results. Consequently, deriving the systematic bias from specular waveform is not very practical in these lakes.

[Figure]

Fig R2 Typical specular waveform (a) and diffuse waveform (b) and average PP values of each cycle in Jason-1 data in the GSL (c).

(3) As for the uncertainty analysis in Section 5.2, It will be modified as suggested by the reviewer.

**Specific comments**

1. Line 83. Why the approach developed in Becker et al., 2017 for SAR waveforms is not compatible with the conventional altimetry waveforms?. SAR waveform is more specular, however the approach remains the same: retracking of the sub-waveforms.

Response: Beckers et al. (2017) used the peak wave power to retrack sub-waveforms, e.g., using the first peak to represent the snow-ice interface and the largest peak to represent the ice-water interface. For specular SAR waveforms, it is easier to locate the peak (Fig. R3(a)). However, for conventional pulse-limited waveforms, the trailing edge is quite noisy such that the waveform peak does not represent any meaningful backscattering surface (Fig. R3 (b)).

On the other hand, the range difference between different peaks is discrete. The mean LIT of the cycle has to be calculated to obtain a continuous result, which is subject to extreme values. With the approach developed by Li et al. (2022), a continuous LIT can be generated from each waveform and a median value can be used to represent the LIT

of the cycle.

[Figure]

[Figure]

Fig. R3 Comparison of CryoSat-2 SAR waveform and Jason-2 pulse-limited waveform on the Great Slave Lake (GSL).

2. Line 107. The word "paradox" is not good here. "Discrepancy? "

Response: The word "paradox" here seems unnecessary, we have removed it.

3. Line 121. I did not find a method based on "combination of the backscatter and waveform". For me it was the combination of LIT retrieved from the waveform and from the backscatter.

Response: The proposed method is not merely a combination of LIT from waveform and backscattering. First, backscattering coefficients were used to determine the ice-on and the ice-off date as used by Zakharova et al. (2021). The waveform-based LIT is first obtained with an established method (Li et al., 2022). Subsequently, waveform-based LIT serves as a reference to calibrate parameters for the logarithmic model and convert backscattering coefficients into LIT. Finally, the waveform-based LIT and the backscatter-based LIT were combined together as the output. The function of the waveform is to provide reference LIT while the function of backscattering coefficients is to determine the ice-covered period and to estimate the LIT for thin ice (< 0.7 m). Therefore, we consider it appropriate to say that the method is based on a combination of backscatters and waveforms.

4. Lines 121-123. I would say "improved" instead of "novel". "Novel" means really a new approach and not modification of function type or threshold value.

Response: We have modified the expression as suggested.

5. Section 2.1. Please, re-write the section providing uniform description of the environmental /climate conditions for each lake. I can't evaluate the severity of the climate comparing the mean annual temperatures with mean July temperatures. The mean (or range) of negative temperatures or T of most cold winter month will tell more in LIT studies.

Response: We have provided a uniform description for the study area, including mean annual temperature, mean monthly temperature in ice seasons, and mean annual precipitation as shown in the table below.

| Lake/region name | Mean air temperature (℃) | Winter Air temperature (℃) | Precipitation (mm) | Location | Reference |
|---|---|---|---|---|---|
| Mackenzie River basin (GBL, GSL, Athabasca Lake) | -10 – 3 | -35 – -25 | 410 | ~115 °W ~62 °N | (Abdul Aziz and Burn 2006; Howell et al. 2009) |
| Baker Lake | -9.6 | -30 – -20 | 157 | 95.28°W 64.13°N | climate.weather.gc.ca and Medeiros et al. (2012) |
| Winnipeg Lake | -0.7 – 1.6 | -20 – -5 | 498 | 97.25°W 52.12°N | climate.weather.gc.ca and Stewardship (2011) |
| Hulun Lake | 2.3 | -16 – -10 | 240 | 117.38°E 48.97°N | (WU Qihui 2019) and (Wang et al., 2017) |
| Har Lake | ~0.8 | -15 – -5 | ~50 | 93.21°E 48.05°N | Estimated from reanalysis data |

6. Lines 140 -142. The sentence is not dedicated to the Study area description.

Response: The sentence has been moved to the Section 2.2.

7. Figure 1. Please, unify the scale for the three small lakes (lower panel) 1 cm = 100 km. In next sections you argue that the Har Lake is small and it was difficult to obtain good estimates due to its size. However, it seems that the Jason - lake cross-section length is quite similar for the Baker and Hur lakes.

Response: We have unified the scale bars in three small lakes. The Jason-lake cross-section in Har Lake is ~20 km while that in Baker Lake is ~40 km. Therefore, we believe that relatively lower performance in Har lake can be explained by the small cross-section.

8.Line 147. only for the ocean topography. Ice sheets studies appeared later.

Response: It has been corrected.

9. Lines 149. earlier studies of Becker et al , 2017 and Duguay et al.,2018 should be cited here as well.

Response: Done.

10. Lines 160-162. SGDR Jason product contains as well the range retrieved with Ice (OCOG) retracker widely used in inland water studies. (see S. Calmant, F.Frappart; S.Biancamaria etc. articles).

Response: We have added the information in the revised manuscript.

11. Lines 169-170. please, check the prepositions in the sentences. Move the information on location of LIT in situ stations into the Table 1. In the Table 1, provide consistent with LIT station coordinates (in decimals). In this lines it would be better to mention that the in situ LIT is observed near the coast, where the ice growth and snow-on-ice conditions can differ from open area of the lakes (especially GSL). See different studies of C.Duguay team for details.

Response: We have moved the station information into Table 1 and emphasized that the in situ data were obtained near the coast as suggested.

12. Table 1. Please summarise here all data used for validation, water level, LIT gauges, model simulation.

Response: We will summarize the data in a table in the revised manuscript.

13. Line 185. Please, give here 2-3 lines about 1-D lake ice model developed by Li et al.,(2022). What does it mean "remote sensing" for this model? What is the accuracy of the LIT simulations for the lakes selected for this study?

Response: The 1-D lake ice model developed by Li et al. (2022) has a similar structure as the HIGHTSI model, but it uses MODIS sensor LST as the upper boundary condition to solve the heat transfer equation within lake ice and surface snow. MODIS albedo was also incorporated to reduce uncertainty in simulated surface snow depth. Therefore, it is referred to as a remote sensing lake ice model. Based on validation against in situ data (e.g., in Baker Lake, GSL, and Pepsi Lake), the remote sensing lake ice model shows an accuracy of 0.1–0.2 m (RMSE). There are no in situ observations available in Hulun Lake and Har Lake. But in general, the lake ice model has better performance for lakes with little or no snow cover, such as Hulun and Har.

14. Line 217. Better to speak "delay-Doppler" or "SAR" altimetry when meaning SC2 or Sentinel-3.

Response: We have revised the expression.

15. SAR altimetry can be seen as the beam-limited only in along-track direction.

Response: We have corrected it in the revised manuscript.

16. Line 220. and in other places : replace wave for waveform.

Response: Done.

17. Line 240. Check c and ci in formulations; provide the reference on c in the ice.

Response: We have corrected the formulation. The speed of light in ice $c_i$ is calculated with $c/n_i$, where $n_i$ is the refractive index of ice at the Ku-band. We have provided the reference on $n_i$ in the revised manuscript (Warren and Brandt 2008).

18. Line 253. Delete "quite"

Response: Done.

19. Line 269. Delete "A possible reason". This is the main reason. Provide a reference.

Response: Done. We have found the reference (Fu and Cazenave 2000; Peureux et al. 2022).

20. Line 272. replace "increases(decreases)" on " is high(low)"

Response: Done.

21. Line 280. Explain STD, variability in spatial domain?

Response: Yes, the STD of backscattering coefficients here represents the variability in the spatial domain. It is calculated with backscattering coefficients obtained in each cycle. We have emphasized this point in the revised manuscript.

22. Line 285. I am not agree. The Sig0 rise during this phase is due to water-on-ice appearance or decreasing of the penetration depth (volume scattering) caused by high water content of the melting snow, unless you prove your statement with the solid references.

Response: The reviewer suggests that high Sig0 in the melting stage (Stage IV) is mainly caused by the presence of meltwater (in snow or on ice surface), while we suggest that this phenomenon is caused mostly by the decrease in LIT and associated decrease in volume scattering and absorption.

A possible piece of evidence could be the timing of such events. For instance, a peak of Sig0 occurred on May 27, 2015 (shown in Fig. 4 in the manuscript) but based on in-situ data the melting started around May 1, 2015 (Fig. R4). The meltwater was likely to occur from the beginning of May, but Sig0 did not reach the peak until the end of May. Therefore, we consider it probably caused by an overall decrease in LIT as opposed to an abrupt change in surface meltwater.

[Figure]

Fig. R4 In-situ LIT of GSL in 2015

23. Line 320. "I1 does not change with LIT" is also only the assumption. If I understood I1 is the surface echo plus snow volume echo. It can change during the winter due to the changes in the snow (densification, redistribution) and ice (thermal or mechanical deformation resulting to changes in ice surface roughness).

Response: Yes, "$I_1$ does not change with LIT" is an assumption to simplify the derivation. We have clarified this point in the revised manuscript and we have also added this point into the Discussion Section.

24. Equation 8. Explain, please, why in the Eq2 "-2kHi" appeared.

Response: The term "-2$kH_i$" in Equation 8 can be derived by substitute $I$ with Equation 6 as shown below. Here $I$ denotes the transmitted microwave intensity that has just reached the bottom of lake ice. $I_0$ denotes the transmitted microwave intensity just below the ice surface. $I_2$ denotes the backscattered intensity from the ice-water interface that is leaving the air-ice interface (the signal has traveled a round-trip in the lake ice).

$$I = I_0 \times e^{-kH_i} \tag{6}$$

$$I_2 = r \times I \times e^{-kH_i} = r \times I_0 \times e^{-2kH_i} \tag{7}$$

$$I_b = I_1 + I_2 = I_1 + r \times I \times e^{-kH_i} = I_1 + r \times I_0 \times e^{-2kH_i} \tag{8}$$

25. Equation 10. Insert space between equations

Response: Done.

26. Line 361 replace the word "paradox"

Response: We have replaced the phrase" with "pending question".

27. Line 369. Please, rephrase the sentence in more scientific manner. Phrase "investigate LSH for ice-covered lakes" is scientific slang.

Response: We have rephrased the sentence as "The LSH for ice-covered lakes were retrieved using different thresholds".

28. Line 374.    All LSH retrievals...

Response: It has been corrected in the revised manuscript.

29. Lines 377-381 These lines describe the results. Move them to the Result section. See my general comment dedicated to the evaluation of the open-water altimetric range bias obtained with 0.1 and 0.5 thresholds.

Response: We have moved this part to Section 4.3.

30. Lines 383. What does it mean "more robust performance"? Please, illustrate with the statistics comparing with corresponding reference time series. Remind in the text what Davis (1997) investigated: inland waters, which retracker (OCOG ) ?.

Response: The reviewer suggests us providing some statistics to show that the 0.5-threshold method has better performance during open water periods compared with the 0.1-threshold method. We calculated statistics for the four lakes as shown in the figure below (Fig. R5). Results show that in each study lake the 0.5-threshold outperforms the 0.1-threshold by 2 – 5 cm in terms of RMSE.

We also provided more descriptions about the reference (Davis, 1997) in the revised manuscript. Davis (1997) developed the threshold retracking method and tested several different thresholds (0.1, 0.2, and 0.5). The conclusion is that for pulse-limited altimeters when the shape of a waveform is dominated by surface scattering (e.g., open water surface), the half-power point represents the mean surface elevation and a 0.5-threshold would be the most appropriate choice.

[Figure]

[Figure]

Fig. R5 Comparison between LSHs based on the 0.1-threshold and the 0.5-threshold during open water periods in four study lakes.

31. Lines 385-387. Not clear how it was implemented: by concatenation of [ LSH0.5 open water; (LSH0.1ice - bias)] ?

Response: Yes, the combined LSH time series is the concatenation of LSHs based the 0.5-threshold during open water periods, and LSHs based on the 0.1-threshold during ice-covered periods with bias removed. We have made it clearer in the revised manuscript.

32. Line 423. The subsection 4.1 was also based on altimetric measurements. Change the title of the subsection 4.2 for more specific.

Response: The title of Section 4.1 has been changed to "LIT based on the combination of waveforms and backscattering coefficients".

33. Line 424. Please, provide here the uncertainties found in the cited study. Not clear why you did not compare your waveform-based retrievals with the in situ observations. Did you use exactly the same areas, same tracks, same codes for selection of the sub-waveforms as in the Li et al., 2022 or you used LIT provided by these authors?

Response: Uncertainty in the waveform-based LIT is 0.15–0.2 m based on the comparison against in situ data (Li et al., 2022). We directly used waveform-based LIT data derived by our previous study, i.e., Li et al. (2022) that is available online at https://doi.org/10.5281/zenodo.5528542.

34. Lines 426-430. Summarise the statistics for each lake in a Table. Be consistent

providing coefficients of correlation or determination.

Response: We have summarized the statistics as shown in the table below.

| Lake name | CC | RMSE (m) | Reference data |
|---|---|---|---|
| Baker Lake | 0.94 | 0.24 | In situ |
| Great Slave Lake | 0.80 | 0.17 | In situ |
| Hulun Lake | 0.94 | 0.11 | Modeled |
| Har Lake | 0.89 | 0.20 | Modeled |

35. Line 432. Why your method (backscatter-based in logarithmic approximation ???) does not depend on availability of in situ observations. If I understood from the lines 339-351 you calibrated the parameters K, A and C ?.

Response: Parameters K, A, and C in the logarithmic model can be calibrated against the waveform-based LIT (see Fig. 3(b) in the manuscript), so in-situ data are not necessary in deriving the backscatter-based LIT. For instance, all the backscatter-based LIT in Section 4.2 were generated in the absence of in situ data. We only used parameters calibrated against in situ data in Section 4.1 to evaluate the feasibility of the logarithmic model and to compare our model with the power function model developed by Zakharova et al. (2021), which is based on parameters calibrated against in situ measurements. We have made this point clearer in the revised manuscript.

In addition, it is worth noting that though the backscatter-based LIT is based on the calibration against waveform based-LIT, they are not functionally identical to each other. The backscattered-based LIT can detect ice thinner than 0.5 m, whereas the waveform-based LIT cannot. Therefore, the backscatter-based LIT can be interpreted as an extrapolation and supplement to the waveform-based LIT.

36. Section 4.2 In the Figure 3 only the equation for GSL was provided. Please, in the section 4.2 give the table with 1) the K, A, C parameters for each lake, 2) with the period used for calibration, 3) period used for validation, 4) accuracy of the LIT for validation period (RMSE, correlation coef.).

Response: The reviewer suggests to provide calibrated model parameters, calibration and validation period, and accuracy for all lakes.

First, for each lake and each winter, the parameter set K, A, and C are different so there are a lot of parameters for each lake. We listed all the parameters in the table below and added it into the Supporting Information.

Second, we used all available waveform-based LIT in each winter to calibrate these parameters, so there is not a period for validation. The reason why we calibrated the parameters in each winter separately is that the relationship between backscattering coefficients and the LIT is highly variable from year to year as a result of multiple factors. Sticking to one parameter set for the entire study period would cause large uncertainties and does not help improve altimetry-based LIT estimation. Therefore, the backscatter-based LIT was meant to be extrapolation and supplement to the waveform-based LIT and should be cautiously used when there is no reference data.

Last, for the accuracy of the calibration period, we provided $R^2$ and RMSE in the table below, showing that for most ice-covered seasons $R^2$ is over 0.7 and the RMSE is smaller than 0.2 m.

| Lake name | Ice season | -1/K | A | C | $R^2$ | RMSE (m) |
|---|---|---|---|---|---|---|
| Great Slave Lake | 2002/11—2003/5 | -0.50 | 1.62 | 21.11 | 0.14 | 0.38 |
| | 2003/11—2004/5 | -0.57 | 1.84 | 17.67 | 0.92 | 0.08 |
| | 2004/11—2005/5 | -1.76 | 6.41 | 0.19 | 0.64 | 0.16 |
| | 2005/11—2006/5 | -1.34 | 4.91 | 0.545 | 0.64 | 0.09 |
| | 2006/11—2007/5 | -1.29 | 5.00 | 0.535 | 0.87 | 0.11 |
| | 2007/11—2008/5 | -1.32 | 4.66 | 3.62 | 0.86 | 0.11 |
| | 2008/11—2009/5 | -1.42 | 5.48 | 0.45 | 0.94 | 0.07 |
| | 2009/11—2010/5 | -1.46 | 5.48 | 0.94 | 0.86 | 0.08 |
| | 2010/11—2011/5 | -1.20 | 4.41 | 5.68 | 0.96 | 0.07 |
| | 2011/11—2012/5 | -1.62 | 6.16 | 0.18 | 0.78 | 0.14 |
| | 2012/11—2013/5 | -1.89 | 6.84 | 0.81 | 0.19 | 0.26 |
| | 2013/11—2014/5 | -1.30 | 4.95 | 0.69 | 0.17 | 0.36 |
| | 2014/11—2015/5 | -1.30 | 4.67 | 0.275 | 0.09 | 0.20 |
| | 2015/11—2016/5 | -1.58 | 5.76 | 0.89 | 0.79 | 0.10 |
| | 2016/11—2017/5 | -1.61 | 6.03 | 0.72 | 0.79 | 0.15 |
| | 2017/11—2018/5 | -0.57 | 1.89 | 14.43 | 0.79 | 0.09 |
| | 2018/11—2019/5 | -1.64 | 6.09 | 0.04 | 0.90 | 0.07 |
| | 2002/11—2003/5 | -1.53 | 5.67 | 5.69 | 0.90 | 0.12 |
| | 2003/11—2004/5 | -0.75 | 2.32 | 12.75 | 0.98 | 0.05 |
| | 2004/11—2005/5 | -2.04 | 7.75 | 0.69 | 0.98 | 0.09 |
| | 2005/11—2006/5 | -1.73 | 6.67 | 2.83 | 0.93 | 0.10 |
| | 2006/11—2007/5 | -0.23 | 1.00 | 18.72 | 0.94 | 0.05 |

| | | | | | |
|---|---|---|---|---|---|
| | 2007/11—2008/5 | -4.21 | 14.89 | 0.23 | 0.26 | 0.40 |
| | 2008/11—2009/5 | -2.18 | 8.18 | 0.27 | 0.96 | 0.12 |
| | 2009/11—2010/5 | -1.30 | 4.23 | 7.05 | 0.96 | 0.07 |
| | 2010/11—2011/5 | -2.27 | 7.80 | 0.6 | 0.85 | 0.18 |
| | 2011/11—2012/5 | -1.80 | 6.78 | 0.33 | 0.48 | 0.31 |
| Baker Lake | 2012/11—2013/5 | -2.13 | 7.54 | 0.045 | 0.87 | 0.15 |
| | 2013/11—2014/5 | -3.62 | 13.01 | 0.2 | 0.75 | 0.21 |
| | 2014/11—2015/5 | -1.66 | 5.46 | 7.56 | 0.69 | 0.24 |
| | 2015/11—2016/5 | -1.30 | 4.57 | 8.39 | 0.91 | 0.15 |
| | 2016/11—2017/5 | -0.66 | 2.17 | 12.46 | 0.96 | 0.06 |
| | 2017/11—2018/5 | -2.16 | 7.97 | 0.97 | 0.96 | 0.10 |
| | 2018/11—2019/5 | -2.53 | 9.12 | 0.3 | 0.97 | 0.08 |
| | 2013/11—2014/5 | -2.72 | 9.96 | 0.87 | 0.81 | 0.12 |
| | 2014/11—2015/5 | -2.44 | 9.48 | 0.55 | 0.67 | 0.09 |
| Hulun Lake | 2015/11—2016/5 | -1.91 | 7.45 | 0.085 | 0.85 | 0.10 |
| | 2016/11—2017/5 | -2.31 | 8.86 | 0.42 | 0.58 | 0.19 |
| | 2017/11—2018/5 | -2.55 | 9.55 | 0.995 | 0.60 | 0.16 |
| | 2018/11—2019/5 | -2.72 | 10.48 | 0.92 | 0.38 | 0.16 |
| | 2005/11—2006/5 | -6.31 | 23.93 | 0.81 | 0.65 | 0.10 |
| | 2006/11—2007/5 | -5.28 | 20.23 | 0.48 | 0.52 | 0.11 |
| | 2007/11—2008/5 | -2.76 | 10.38 | 14.28 | 0.69 | 0.08 |
| | 2008/11—2009/5 | -0.51 | 1.59 | 33.78 | 0.96 | 0.04 |
| | 2009/11—2010/5 | -1.04 | 3.55 | 19.92 | 0.89 | 0.09 |
| Har Lake | 2010/11—2011/5 | -0.59 | 1.73 | 34.21 | 0.77 | 0.16 |
| | 2011/11—2012/5 | -0.47 | 1.19 | 31.45 | 0.84 | 0.09 |
| | 2012/11—2013/5 | -6.35 | 24.59 | 0.87 | 0.90 | 0.08 |
| | 2013/11—2014/5 | -0.37 | 1.09 | 34.26 | 0.77 | 0.09 |
| | 2014/11—2015/5 | -2.19 | 8.50 | 6.84 | 0.98 | 0.03 |
| | 2015/11—2016/5 | -1.38 | 4.36 | 21.84 | 0.94 | 0.03 |
| | 2016/11—2017/5 | -4.97 | 18.82 | 0.48 | 0.1 | 0.14 |
| | 2017/11—2018/5 | -3.38 | 13.43 | 0.11 | 0.92 | 0.06 |
| | 2018/11—2019/5 | -2.98 | 11.91 | 0.64 | 0.92 | 0.08 |

37. Lines 440-445. Some reasoning based on demonstrations via figures or tables of statistics is necessary to prove why waveform-retrieved LIT was used for thick ice, while the backscatter-based LIT retrievals were used for thin ice range.

Response: To better show that the backscatter-based LTI is suitable for thin ice while the waveform-based LIT is efficient for thick ice, we have conducted another test in a European lake named Pepsi. The maximum ice thickness in Pepsi lake is ~ 0.5 m and the total thickness of snow and ice is 0.6–0.9 m, much smaller than the four lakes we tested in the manuscript.

Results in Fig R6 clearly show that waveform-based LIT can only detect lake ice and snow over 0.5 m, while the backscattered LIT can reflect the initial stage of ice accumulation. On the other hand, as we mentioned in Section 4.1, backscattering coefficients have some saturation effect when the ice is very thick, resulting in underestimation of thick ice. Therefore, it is a straightforward idea to combine their advantages and reduce their disadvantages by using the backscattered LIT for thin ice and waveform-based LIT for thick ice. In addition, the backscattered LIT in Pepsi lake is derived with parameters calibrated against waveform-based LIT, showing that even limited waveform-based LIT measurements can be valuable to convert backscattering coefficients into LIT.

[Figure]

Fig R6 Comparison between backscattered-LIT, waveform-based LIT, and in situ snow and ice thickness in Pepsi Lake.

38. Line 453. For me, the track length within lakes Baker and Har looks the same (Figure 1), so the reason given in the text is not strong.

Response: The track length within Har lake is ~20 km while that in Baker Lake is ~40 km. In addition, the track length of Hulun lake is ~ 30 km. Given the similar climate condition and geolocation, relatively lower performance in Har lake than that in Hulun Lake is probably caused by the smaller cross-section.

39. Figure 9. For simulated LIT+SnowDepth provide the line as well. The gray shadow-black line difference is not visible.

Response: We have revised the figure as suggested.

40. Line 466. What the "effective water level" is? Hydrostatic water level or water-ice interface?

Response: Here the "effective water level" denotes the Hydrostatic water level. We have replaced the phrase to make it clearer.

41. Line 468. improvement comparing to what?. I prefer to see RMSE for each lake compared to the RMSE of "what this improvement refers to". Please, give the metrics found in Yang. etal 2022.

Response: The improvement here means that the merged LSH time series outperforms those based solely on either the 0.1 or 0.5 threshold method. We provided metrics for each lake in the table below. Results show that the merged LSH outperforms the 0.1-threshold method by 0.9–1.7 cm, and the 0.5-threshold method by 2.5–9.2 cm.

Yang et al. (2021) used the 0.1-threshold in three lakes (GSL, GBL, and Athabasca Lake) and found metrics between 0.06–0.12 m in terms of standard deviation, similar to ours. However, their results were derived based on smoothed waveforms (moving average filter), different sub-waveform selection strategy, and different tracks and gauge stations, which could affect the outcome considerably. Therefore, we did not directly compare our merged LSH time series with their results, but with our own LSH time series based on 0.1-threshold and 0.5-threshold.

| Lake name | Merged LSH RMSE (cm) | 0.1-threshold RMSE (cm) | 0.5-threshold RMSE (cm) |
| --- | --- | --- | --- |
| GSL | 7.1 | 8.0 | 14 |
| GBL | 8.1 | 9.4 | 10.6 |
| Athabasca Lake | 9.8 | 11.5 | 12.5 |
| Winnipeg Lake | 10.2 | 11.7 | 19.4 |

42. Figure 10. and everywhere. STD and RMSE statistics should be round to centimetre.

The accuracy of the altimetry height retrievals over the inland water objects is still from several centimetres - to several tens of centimetres.

Response: We have updated the metrics as suggested.

43. Lines 480-485. For me, the variability of the RMSE and STD between the lakes is low. Moreover, the STD is almost equal to RMSE, so keep only the RMSE. The explication of observed inter-lake uncertainties provided in this paragraph is unrealistic. The text does not correspond to the figure.

Response: We only keep RMSE in the revised manuscript as suggested, and have reduced the explication of observed inter-lake uncertainties in the revised manuscript.

44. For the Discussion Section see my general comments.

Response: We have addressed the issue related to the Discussion section in our response to the general comments.

References

Abdul Aziz, O.I., & Burn, D.H. (2006). Trends and variability in the hydrological regime of the Mackenzie River Basin. *Journal of Hydrology, 319*, 282-294

Fu, L.-L., & Cazenave, A. (2000). *Satellite altimetry and earth sciences: a handbook of techniques and applications*. Elsevier

Howell, S.E.L., Brown, L.C., Kang, K.-K., & Duguay, C.R. (2009). Variability in ice phenology on Great Bear Lake and Great Slave Lake, Northwest Territories, Canada, from SeaWinds/QuikSCAT: 2000–2006. *Remote Sensing of Environment, 113*, 816-834

Medeiros, A.S., Friel, C.E., Finkelstein, S.A., & Quinlan, R. (2012). A high resolution multi-proxy record of pronounced recent environmental change at Baker Lake, Nunavut. *Journal of Paleolimnology, 47*, 661-676

Peureux, C., Longépé, N., Mouche, A., Tison, C., Tourain, C., Lachiver, J.m., & Hauser,

D. (2022). Sea-ice detection from near-nadir Ku-band echoes from CFOSAT/SWIM

scatterometer. *Earth and Space Science, 9*, e2021EA002046

Stewardship, M.W. (2011). State of Lake Winnipeg: 1999 to 2007. In: Environment Canada and Manitoba Water Stewardship

Warren, S.G., & Brandt, R.E. (2008). Optical constants of ice from the ultraviolet to the microwave: A revised compilation. *Journal of Geophysical Research: Atmospheres, 113*

WU Qihui, L.C., SUN Biao, SHI Xiaohong, ZHAO Shengnan, HAN Zhiming (2019). Change of ice phenology in the Hulun Lake from 1986 to 2017. *PROGRESS IN GEOGRAPHY, 38*, 1933-1943

WANG Jingjie LI Changyou SUN Biao FAN Cairui LIANG Lie HAN Zhiming (2017) Impacts of Precipitation on Runoff Yield of Hulun Lake Basin During 1963-2014. *Bulletin of Soil and Water Conservation*, 37(2), 115-119

---

## Author Comment (AC2)

Response Letter

Reviewer Comments 2

Lake ice is a very important component of the cryosphere, and couples closely with global warming and local climate. This manuscript provided a detailed method on estimating ice thickness and water level for ice-covered lakes. Some field measurements were also included to test the feasibility of the method. I think the main problems currently are the lack of some detailed explanations on method itself and also on the results, as listed below.

Response: Thanks for all these insightful and constructive comments. We have made tremendous effort to improve the clarity and consistency in the Method and Result sections and provided more discussion related to snow impact on the method as suggested by the Reviewer. Specific comments are addressed point-by-point in the following context.

1.  A table summarizing all seven lakes are possible more direct to readers to understand them, except for Figure 1. And what are the red numbers in Figure 1?

Response: We have summarized related information on lakes studied in the table below. Red lines in Figure 1 represent Jason-1/2/3 ground tracks and numbers denote the track number.

| Lake/region name | Mean air temperature (°C) | Winter Air temperature (°C) | Precipitation (mm) | Location | Reference |
|---|---|---|---|---|---|
| Mackenzie River basin (GBL, GSL, Athabasca Lake) | -10 – 3 | -35 – -25 | 410 | ~115 °W ~62 °N | (Abdul Aziz and Burn 2006; Howell et al. 2009) |
| Baker Lake | -9.6 | -30 – -20 | 157 | 95.28°W 64.13°N | climate.weather.gc.ca and Medeiros et al. (2012) |
| Winnipeg Lake | -0.7 – 1.6 | -20 – -5 | 498 | 97.25°W 52.12°N | climate.weather.gc.ca and Stewardship (2011) |
| Hulun Lake | 2.3 | -16 – -10 | 240 | 117.38°E 48.97°N | (WU Qihui 2019) and (Wang et al., 2017) |

| | | | | | |
|---|---|---|---|---|---|
| Har Lake | ~0.8 | -15 – -5 | ~50 | 93.21°E  48.05°N | Estimated from reanalysis data |

2.  L147-149 can be moved to the introduction section.

Response: Done.

3.  There no citation to Figure 3b. And the text on Figure 3a seems not to match the explanations on L 215-225.

Response: Figure 3b was cited in L347. We have revised Figure 3(a) to make it consistent with the method description.

4.  L285ï¼Œ"the highest peak in the freezing period and the highest peak in the melting period were chosen to characterize the ice-on and ice-off dates ". It is easy to validate these dates. Do you have some validations to the field measurements? In lake ice cycle, the ice-on date and ice-off date are always not at a single day, instead the process would last for several days sometimes. But there is an obvious peak according to Figure 4, how to correspond to the real conditions in lake ice?

Response: The reviewer suggests us validating ice-on and ice-off dates identified from backscattering coefficients. The reviewer also questions how peaks in backscattering coefficients are related to real conditions in lake ice, which (based on our understanding) is characterized by events such as freeze-up start (FUS), freeze-up end (FUE), break-up start (BUS), and break-up end (BUE).

We did not find in-situ lake ice phenology data for lakes in this study, so we compared derived ice-on and ice-off dates with optical images in GSL. FUS, FUE, BUS, and BUE dates were manually identified from MODIS images from 2009 to 2016 and compared with backscatter-based ice phenology. Results shown in the figure below suggest that backscatter-based ice-on dates are very close to the FUE date (RMSE = 3 days), while backscatter-based ice-off dates are close to the BUE date (RMSE = 9 days). Given the 10-day repeat cycle of Jason-1/2/3 altimeters, the overall performance of backscatter-based ice phenology is satisfactory.

[Figure]

Fig R7 Comparison between the backscatter-based and the MODIS-based lake ice phenology in GSL during 2009 to 2016. FUS, FUE, BUS, and BUE denote freeze-up start, freeze-up end, break-up start, and break-up end of lake ice, respectively.

5. What is Sig on L293? Do we have a mathematical expression on function CumSum?

Response: *Sig* here represents backscattering coefficients. We have made it clear in the revised manuscript. In addition, the expression of CumSum can be written as:

$$CumSum(x_n) = \sum_{i=1}^{n} x_i$$

6. L303,"we can derive LITs based on backscattering coefficients without in situ ice thickness measurements. "I don't understand this sentence. There is not necessary to validate results of remote sensing?

Response: "we can derive LITs based on backscattering coefficients without in situ ice thickness measurements" suggests that our method can be applied to lakes without in situ data to give an initial estimation of LIT. The reason why our method does not rely on in situ information to retrieve LIT has been explained in detail in our response to Comment 35 from Reviewer 1. However, to validate the remote sensing results we still need in-situ measured LIT and snow depth as reference data.

7. L303-305. Both equation (10) and section 4.1 are cited here, then can we put these sentences in later sections?

Response: We cite Equation (10) and Section 4.1 here to help readers find core results for the application so that they do not have to go through the derivation part if not necessary.

8.    The reflection on air-snow interface is not shown on Figure 5, and also not in the equation 5. A schematic diagram like Figure 5 should be placed in the front of the method section.

Response: In original Figure 5 and Equation 5, $I_1$ denotes the backscattered intensity from the air-snow interface and the snow-ice interface (L310). Therefore, the snow thickness was implicitly included in the derivation of the method.

However, to avoid confusion, we have used different symbols to represent the two paths in the revised manuscript as shown in the figure below. $I_0$ now denotes the incident intensity at the air-snow interface, $I_1$ denotes the backscattered intensity from the air-snow interface, $I_2$ denotes the backscattered intensity from the snow-ice interface, and $I_3$ denotes the backscattered intensity from the ice-water interface. In addition, we have moved Figure 5 to the front of the Method Section as suggested.

[Figure]

Fig R8 Modified Figure 5

9.    Equation 6, Hi should be the ice thickness, not the thickness of snow and ice if according to Figure 5.

Response: We have revised this part. Now $H_i$ denotes the lake ice thickness, and $H_s$ denotes the lake snow depth. The revised derivation of Equations 5–11 is detailed in our response to Comment 10 of Reviewer 2.

10. In equations 5-11, there are some key points not mentioned. First, snow thickness cannot be ignored because it was even larger than the ice thickness on some boreal lakes. Secondly, main reflections occur on the air-snow interface rather than the snow-ice interface, the attenuation in the snow layer also cannot be ignored. Overall, snow is a key impact factors on the lake ice remotes sensing, do we have some discussions on this issue, especially for equations 5-11?

Response: This reviewer suggests: (1) considering the snow thickness in the derivation of Equations 5–11 and (2) providing a discussion on the reflection and attenuation of the radar pulse in snow layers.

We did not ignore lake surface snow. In fact, we have suggested in the manuscript that the backscattered intensity from the upper layer could come from the snow-air interface (L 319) and the derived backscattered-LIT and waveform-based LIT is close to the total thickness of ice and snow. To make this point clearer, we have revised this part as suggested by the reviewer. But the core results (the logarithmic regression model) did not change.

To simplify this analysis, we assumed an effective extinction coefficient $k$ for both snow and ice layers. As suggested by the Reviewer, backscattering from the snow-ice interface is very small, so we used a constant to approximate the backscattered intensity from the air-snow interface and snow-ice interface. In the revised manuscript, Equations 5–11 are now written as:

$$I_b = I_1 + I_2 + I_3 = R_1 I_0 + I_2 + I_3 \tag{5}$$

$$I_2 = R_2(1 - R_1)I_0 \times e^{-2kH_s} \tag{6}$$

$$I_3 = R_3(1 - R_2)(1 - R_1)I_0 \times e^{-2k(H_s + H_i)} \tag{7}$$

$$I_b = (R_1 I_0 + I_2) + R_3(1 - R_2)(1 - R_1)I_0 \times e^{-2k(H_s + H_i)} \tag{8}$$

$$\sigma_0 = A + B \times e^{-K(H_s + H_i)} \tag{9}$$

$$(H_s + H_i) = -\frac{1}{K} \times \ln(\sigma_0 - A) + C, C = \frac{\ln(B)}{K} \tag{10}$$

$$\sigma_0 = A + \sigma_{max} \times e^{-K(H_s + H_i)} \tag{11}$$

where $I_0$, $I_1$, $I_2$, and $I_3$ denote the incident intensity at the air-snow interface, the backscattered intensity from the air-snow interface, the backscattered intensity from the snow-ice interface, and the backscattered intensity from the ice-water interface. As shown in revised Fig. 5., $R_1$, $R_2$, and $R_3$ denote the reflectance from air-snow, snow-ice, and ice-water interfaces. $I_b$ is total backscattered intensity, $H_s$ is the snow depth, $H_i$ is the ice thickness, $k$ is the effective extinction coefficient of lake ice and snow, and $\sigma_0$ is the backscattering coefficient. A, C, and K are model parameters to be calibrated.

As for the impact of snow on lake ice remote sensing, please see our response to Comment 13 of Reviewer 2.

11. There are some obvious unusual points on Figure 7a, do we have some explanations here?

Response: The outliers marked by orange circles in Figure 7(a) (shown below) are caused mostly by the initial condition of LIT and the nature of the power function model (Zakharova et al., 2021). As we mentioned earlier, the Jason-detected ice-on date is close to the FUE when thin lake ice has completely covered the lake surface. Based on the power function model, on the ice-on date detected by Jason-1/2/3, the initial backscattered LIT equals zero, but the real LIT could be several decimeters. Therefore, the outliers are located on the y-axis.

[Figure]

Figure 7(a) Scatter plot of in situ LIT and backscattered LIT (power function model) in GSL

12. Figures 8-9. These lakes belongs to totally different climate regions. The snow and ice thickness of Baker Lake and GSL are larger than that of Hulun Lake and Har Lake. Thicker snow will introduce more uncertainty into remote sensing as the author said on L446-448, but the results in Figure 8 seems to be better than that in Figure 9. And why CC was employed in Figure 8 while $R^2$ was in Figure 9?

Response: We have unified the metrics in Figures 8 and 9. We also provided a table to summarize the metrics of the four lakes as shown below. The CC of Hulun Lake is 0.94, similar to Baker Lake, while the CC of Har Lake is 0.89, higher than GSL. The method has the best performance in Hulun Lake, consistent with the argument that the method is more efficient in lakes with less snow cover. As for Har lake, the lower performance is probably caused by the small cross-section of the lake as we mentioned in the manuscript (L 452).

| Lake name | CC | RMSE (m) | Reference data |
|-----------|------|----------|----------------|
| Baker Lake | 0.94 | 0.24 | In situ |
| Great Slave Lake | 0.80 | 0.17 | In situ |
| Hulun Lake | 0.94 | 0.11 | Modeled |
| Har Lake | 0.89 | 0.20 | Modeled |

13. The radar backscattering coefficient depends on the ice crystal type such as granular ice and columnar ice, and also on the gas bubble size, ice salinity et al. The information on the ice physics are not mentioned here, and will the difference in ice physics of these lakes pose some impacts on the thresholding process?

Response: Ice and snow physics is very important for understanding observed microwave signals in lake ice. Several studies have discussed the lake ice and snow physics and their impacts on SAR imaging systems which has large incident angle compared to altimetry systems (Atwood et al. 2015; Gherboudj et al. 2009; Han and Lee 2012; Tjuatja et al. 1992). Here we try to provide an overall description of possible impacts of the snow and ice physics based on the literature.

Regarding backscattering coefficients, surface scattering is affected by the roughness and dielectric constant ($\varepsilon$) of the snow/ice surface; volume scattering is caused by snow particles and air bubbles captured inside ice, while ice-bottom scattering is controlled mostly by the roughness and $\varepsilon$ of the ice/water interface. Snow cover can increase backscattering coefficients of Ku-band radar obtained from frozen lakes (Gunn et al. 2015). Based on Kim et al. (1984), thicker snow cover contributes more to backscattering coefficients due to enhanced volume scattering. Among the rest backscattering sources, lake ice bubbles were initially regarded as an important factor. However, Atwood et al. (2015) show that backscattering from ice bubbles is insignificant in terms of magnitude compared with reflectance from the ice-water interface. Therefore, the roughness of the ice-water interface could be a critical factor that controls the backscattered intensity.

We can conclude that backscattering coefficients obtained from thick snow-covered lakes correspond to more information from the snow layer and less information from the ice layer, contributing to larger uncertainty in these lakes. On the other hand, the roughness of the ice-water interface has a large influence on the backscattered intensity, which could be the reason why the relationship between LIT and backscattering coefficients varies considerably from year to year.

For the waveform-based LIT, the most important physical property is the $\varepsilon$ of snow and ice, as it determines the speed of light within snow and ice and the timing of reflected

signals from different interfaces (higher $\varepsilon$ corresponds to the lower speed of light). During the ice accumulation process, the $\varepsilon$ of ice is relatively stable. The $\varepsilon$ of dry snow is almost solely dependent on the snow density (Tiuri et al. 1984), which can be approximated with $\varepsilon = 1 + 2\rho$, where $\rho$ is the relative snow density (with respect to water).

We used the same constant $\varepsilon$ for both ice and snow, which is a compromise as we do not have any prior information related to snow depth and density. Because the waveform-based method measures the time difference between different interfaces, at the beginning of ice and snow accumulation, our method could slightly underestimate the total thickness of snow and ice because snow has a smaller $\varepsilon$ and a larger speed of light. As the snow becomes denser during the frozen period and the speed of light becomes slower in snow, the waveform-based LIT could be closer to the total thickness of snow and ice.

References

Abdul Aziz, O.I., & Burn, D.H. (2006). Trends and variability in the hydrological regime of the Mackenzie River Basin. *Journal of Hydrology, 319*, 282-294

Atwood, D.K., Gunn, G.E., Roussi, C., Wu, J., Duguay, C., & Sarabandi, K. (2015). Microwave backscatter from Arctic lake ice and polarimetric implications. *IEEE Transactions on Geoscience and Remote Sensing, 53*, 5972-5982

Gherboudj, I., Bernier, M., & Leconte, R. (2009). A backscatter modeling for river ice: Analysis and numerical results. *IEEE Transactions on Geoscience and Remote Sensing, 48*, 1788-1798

Gunn, G.E., Brogioni, M., Duguay, C., Macelloni, G., Kasurak, A., & King, J. (2015). Observation and modeling of X-and Ku-band backscatter of snow-covered freshwater lake ice. *IEEE Journal of Selected Topics in Applied Earth Observations and Remote Sensing, 8*, 3629-3642

Han, H., & Lee, H. (2012). Radar backscattering of lake ice during freezing and thawing stages estimated by ground-based scatterometer experiment and inversion from genetic algorithm. *IEEE Transactions on Geoscience and Remote Sensing, 51*, 3089-3096

Howell, S.E.L., Brown, L.C., Kang, K.-K., & Duguay, C.R. (2009). Variability in ice phenology on Great Bear Lake and Great Slave Lake, Northwest Territories, Canada, from SeaWinds/QuikSCAT: 2000–2006. *Remote Sensing of Environment, 113*, 816-834

Kim, Y.-S., Onstott, R., & Moore, R. (1984). Effect of a snow cover on microwave backscatter from sea ice. *IEEE Journal of oceanic Engineering, 9*, 383-388

Medeiros, A.S., Friel, C.E., Finkelstein, S.A., & Quinlan, R. (2012). A high resolution multi-proxy record of pronounced recent environmental change at Baker Lake, Nunavut. *Journal of Paleolimnology, 47*, 661-676

Stewardship, M.W. (2011). State of Lake Winnipeg: 1999 to 2007. In: Environment

Canada and Manitoba Water Stewardship

Tiuri, M., Sihvola, A., Nyfors, E., & Hallikaiken, M. (1984). The complex dielectric constant of snow at microwave frequencies. *IEEE Journal of oceanic Engineering, 9*, 377-382

Tjuatja, S., Fung, A.K., & Bredow, J. (1992). A scattering model for snow-covered sea ice. *IEEE Transactions on Geoscience and Remote Sensing, 30*, 804-810

WU Qihui, L.C., SUN Biao, SHI Xiaohong, ZHAO Shengnan, HAN Zhiming (2019). Change of ice phenology in the Hulun Lake from 1986 to 2017. *PROGRESS IN GEOGRAPHY, 38*, 1933-1943

WANG Jingjie LI Changyou SUN Biao FAN Cairui LIANG Lie HAN Zhiming (2017) Impacts of Precipitation on Runoff Yield of Hulun Lake Basin During 1963-2014. *Bulletin of Soil and Water Conservation*, 37(2), 115-119